



Hydrology and
Earth System
Sciences

# On the potential of variational calibration for a fully distributed hydrological model: application on a Mediterranean catchment

**Maxime Jay-Allemand**[1,2], **Pierre Javelle**[1], **Igor Gejadze**[2], **Patrick Arnaud**[1], **Pierre-Olivier Malaterre**[2], **Jean-Alain Fine**[3], **and Didier Organde**[3]

[1]RECOVER, INRAE, Aix-Marseille Université, 3275 Route de Cézanne, 13182 Aix-en-Provence, France
[2]INRAE, UMR G-EAU, 361 rue Jean-François Breton, 34196 Montpellier, France
[3]HYDRIS Hydrologie, Parc Scientifique Agropolis II, 2196 Boulevard de la Lironde, 34980 Montferrier-sur-Lez, France

**Correspondence:** Maxime Jay-Allemand (maxime.jay-allemand@irstea.fr)

**Abstract.** Calibration of a conceptual distributed model is challenging due to a number of reasons, which include fundamental (model adequacy and identifiability) and algorithmic (e.g., local search vs. global search) issues. The aim of the presented study is to investigate the potential of the variational approach for calibrating a simple continuous hydrological model (GRD; Génie Rural distributed involved in several flash flood modeling applications. This model is defined on a rectangular $1 \, km^2$ resolution grid, with three parameters being associated with each cell. The Gardon d'Anduze watershed ($543 \, km^2$) is chosen as the study benchmark. For this watershed, the discharge observations at five gauging stations, gridded rainfall and potential-evapotranspiration estimates are continuously available for the 2007–2018 period at an hourly time step.

In the variational approach one looks for the optimal solution by minimizing the standard quadratic cost function, which penalizes the misfit between the observed and predicted values, under some additional a priori constraints. The cost function gradient is efficiently computed using the adjoint model. In numerical experiments, the benefits of using the distributed against the uniform calibration are measured in terms of the model predictive performance, in temporal, spatial and spatiotemporal validation, both globally and for particular flood events. Overall, distributed calibration shows encouraging results, providing better model predictions and relevant spatial distribution of some parameters. The numerical stability analysis has been performed to understand the impact of different factors on the calibration quality. This analysis indicates the possible directions for future develop-

ments, which may include considering a non-Gaussian likelihood and upgrading the model structure.

## 1 Introduction

Flash flood prediction remains a challenging task of modern hydrology due to a number of reasons. First, the heavy precipitation events (HPEs) leading to flash floods are difficult to forecast due to complexity of the processes involved (deep convection triggered by orographic lifting, low-level wind convergence and/or cold pools) (Ducrocq et al., 2016). Second, the hydrological response of the watershed is difficult to model, since it depends on many factors. These include the watershed properties (topography, geology and land cover) and its initial state, for example the soil moisture (Braud et al., 2016). For the western Mediterranean region, which is often affected by flash floods, the HyMeX program (Hydrological Cycle in the Mediterranean Experiment) offers a good opportunity to conduct multidisciplinary studies on the relevant subjects (Drobinski et al., 2014).

In order to better predict flash floods and reduce their potentially devastating impact, warning systems have been developed or are currently under development (Collier, 2007; Hapuarachchi et al., 2011; Gourley et al., 2017). The distributed hydrological models utilizing the rainfall radar measurements are widely implemented in such systems. These models take into account the spatial variability of the catchment properties and of the rainfall and are capable of predicting the discharge at ungauged locations. The latter is

important for small- or medium-sized watersheds which are not covered by an extensive gauging network (Borga et al., 2011). Among operational models presently used for flash flood prediction at a national scale, one could mention the CREST (Coupled Routing and Excess Storage) model in the USA developed by Wang et al. (2011) or the G2G (Grid-to-Grid) model (UK) from Bell et al. (2007). Those distributed models are often "conceptual", because considering more complex "physically based" models may not be justified for the flash flood prediction purpose (Beven, 1989). Since the conceptual parameters are not directly observable, they have to be defined using calibration. However, due to a potentially significant number of cells or subcatchments, the calibration process has to deal with overparameterization and uniqueness (equifinality) issues (Beven, 1993, 2001). Another set of difficulties stems from a dubious adequacy of such models, in which case the very definitions of "overparameterization" and "overfitting" should be refined.

As noticed in De Lavenne et al. (2019), all existing calibration methods developed for the distributed hydrological models involve some sort of regularization strategy. One possible approach is the control set reduction. For example, for each distributed parameter one can try to evaluate a nonuniform spatial pattern from information about the catchment characteristics, including its geological formation, soil properties and land use (Anderson et al., 2006). Then, instead of calibrating the local values of parameters associated with each grid cell, one calibrates a few "superparameters" (additive constant, multiplier and power) that modify this pattern according to a chosen law (see, e.g., Pokhrel and Gupta, 2010). The same idea but in the multiscale setting is implemented in the multiscale parameter regionalization (MPR) method described in Samaniego et al. (2010). Other strategies can also include the use of additional data, as in Rakovec et al. (2016), where a satellite-based total water storage (TWS) anomaly is used to complement the discharge data. For a low-dimensional unknown vector one can use a variety of probabilistic or gradient-free methods to find the sought estimate. It has to be recognized, however, that evaluating useful spatial patterns from auxiliary information is a difficult task by itself. In the presented paper we investigate the possibility of calibrating the distributed parameters without considering any predefined spatial structures. Such a calibration problem falls into the category of high-dimensional inverse problems which can be addressed through the appropriate data assimilation methods.

Methods of data assimilation (DA) have been engaged for several decades in geosciences, including meteorology, oceanography, river hydraulics and hydrology applications. These methods are used for estimating the driving conditions, states and/or parameters (calibration) of a dynamical model describing the evolution of natural phenomena. The estimates are conditioned on observations (usually incomplete) of a prototype system. Some early applications of DA in hydrology are described in the review paper of McLaugh-

lin (1995). More recently the review paper of Liu et al. (2012) reports the progress and challenges of data assimilation applications in operational hydrological forecasting. It seems that the Kalman filtering has been recently the most popular DA method in hydrology (Sun et al., 2016). For instance, in Quesney et al. (2000) the extended Kalman filter is applied with a lumped conceptual rainfall–runoff model to estimate the soil moisture by assimilating the SAR (synthetic aperture radar) data. In Munier et al. (2014), the standard Kalman filter is applied with the semi-distributed conceptual TGR (Transfer with the Génie Rural) model, where the discharge observations are assimilated to adjust the initial model states. It has been shown that the predictive performance depends on the degree of "spatialization" of the watershed and on the number of gauging stations engaged. In Sun et al. (2015), the extended Kalman filter is used with the distributed SWAT (soil and water assessment tool) model to improve flood prediction on the upstream Senegal River catchment. In this work, given the large number of state variables, only the spatially averaged low-resolution updates are estimated. This shows that for DA involving distributed models, scalable methods must be used (scalable algorithm is the one able to maintain the same efficiency when the workload grows). The choice of DA methods is, therefore, limited to the ensemble Kalman filter and the variational estimation.

In variational estimation, one looks for the minimum of the cost function dependent on the control vector (i.e., vector of unknown model inputs) using a gradient-based iterative process. The cost function itself represents the maximum a posteriori (MAP) estimator, which turns into the standard 4D-Var (variational) cost function (Rabier and Courtier, 1992) under the Gaussian assumption. The key issue of variational estimation is the method used for computing the gradient. For a low-dimensional control vector the finite-difference approach can be used. For example, Abbaris et al. (2014) explored such a variational-estimation algorithm involving the lumped conceptual HBV (Hydrologiska Byråns Vattenbalansavdelning) model in operational setting. It has been used to update the soil moisture and the states of the routing tank reservoirs on some events. It has been shown that DA helps to improve the peak flow prediction; however the correct choice of the assimilation period and the forecast horizon is vital. In Thirel et al. (2010), the cost function is minimized iteratively using the BLUE (best linear unbiased estimator) formulation, which is equivalent to the "algebraic" form of the Gauss–Newton method. Here, DA is implemented involving the SIM (SAFRAN–ISBA–MODCOU; Système d'analyse fournissant des renseignements atmosphériques à la neige–Interaction Sol-Biosphère-Atmosphère–MODlisation COUplée) model. It has been shown that the improved estimate of the moisture of the soil layers leads to a significantly better discharge simulation. However, the genuine variational-estimation method relies on the adjoint model, which allows the precise (up to round-off errors) gradient of the cost function to be computed in a single adjoint run. Then,

different minimization methods can be applied. For example, in weather and ocean forecasting, where the models involved are computationally very expensive, the Gauss–Newton method (e.g., "incremental approach") is used. This method (as any local-search method) leads to a nearest local minimum in the vicinity of the prior guess. This could be a serious problem if the posterior distribution is multimodal. Certain past attempts with the local-search methods in hydrology were not always successful, and several authors have reported that these methods fail to deliver the global optimal solution (Moradkhani and Sorooshian, 2009; Abbaspour et al., 2007). For high-dimensional but relatively inexpensive models, the gradient-enhanced global-search minimization methods can be considered (Laurent et al., 2019).

Using the variational estimation involving the adjoint is very common in atmospheric and oceanographic applications. But, in hydrology, only a very few cases have been actually reported. In particular, in Castaings et al. (2009) and Nguyen et al. (2016), the adjoint model has been generated or derived for the kinematic-wave overland flow model with the source term including the rainfall as a driving condition and the infiltration term described by a Green–Ampt model (Castaings et al., 2009) or the Horton model (Nguyen et al., 2016). Since models are represented by a partial differential equation, this is a standard case for which a significant experience has been accumulated within the data assimilation community. The major difference between the two papers is that in Castaings et al. (2009) the adjoint has been generated by automatic differentiation applied to the existing MARINE (Modélisation de l'Anticipation du Ruissellement et des Inondations pour des évéNements Extrêmes) model, while in Nguyen et al. (2016) it is derived and implemented manually. In Castaings et al. (2009) the distributed parameters of the infiltration model has been calibrated considering a single flood event in an identical twin experiment framework, whereas in Nguyen et al. (2016) the author looks for a few global parameters considering two realistic events. In Seo et al. (2009) the adjoint is used for state updating of a lumped model, while in Lee et al. (2012) it is for state updating of a distributed model.

The aim of our study is to assess a set of parameters which may represent the spatially varying hydrological properties of a chosen watershed; thus the distributed model parameters have to be calibrated over a very long assimilation window (i.e., several years). With this purpose we develop a variational-calibration method using the adjoint applied on a simple fully distributed model (GRD; Génie Rural distributed), involving a conceptual cell-to-cell routing scheme. This scheme has been designed keeping in mind the differentiability requirement. The adjoint is obtained by automatic differentiation and manually optimized to provide the capacity to work for long time periods (up to several consecutive years) over large spatial areas, with fine resolution. This requires a memory efficient and fast code. The distributed parameters of the GRD model are calibrated over a French Mediterranean catchment, the Gardon d'Anduze, using rainfall radar data and the discharge data from the outlet gauge station. The discharge data from other gauge stations available in this catchment are used for cross-validation (10 years, being split in two periods). Thus, the major questions addressed in this paper are: (a) can we, in principle, benefit from considering the spatially distributed set of coefficients given by the method instead of the uniform (homogeneous) set of coefficients, and, if so, to what extent? In particular, does it help to improve the discharge prediction over the catchment area including "ungauged" locations? (b) What are the major difficulties associated with this approach (insufficient data, structural deficiency of the model, identifiability issue, etc.), and what could possibly be done to improve the model predictive performance?

The paper is organized as follows. In Sect. 2.1 the GRD model used in this study is described. In Sect. 2.2 we present the variational-estimation algorithm adapted for the parameter calibration purpose. The testing benchmark is described in Sect. 2.3, and the testing methodology in Sect. 2.4. The results are presented in Sect. 3, followed by the "Discussion and conclusions" section.

## 2 Methodology and data

### 2.1 Distributed rainfall–runoff model GRD

The GRD model is a conceptual distributed hydrological model designed for flash flood prediction (Javelle et al., 2010, 2016; Arnaud et al., 2011; Javelle et al., 2014). Since March 2017, it has been used operationally by the national French flash flood warning system called "Vigicrues Flash" (Javelle et al., 2019). The model version used in the present study has been specially developed for testing the potential of distributed calibration using the variational approach. It is defined on a regular $1 \, \text{km}^2$ grid and runs continuously at an hourly time step. For each time step the model input includes the gridded rainfall and potential evapotranspiration, and the output is the discharge field defined at the routing scheme nodes.

Our model incorporates some features from the GR (Génie Rural) model family, which include several lumped and semi-distributed bucket-style continuous models developed over the last 30 years at INRAE Antony (Institut national de recherche pour l'agriculture, l'alimentation et l'environnement). Those models have been extensively tested and have demonstrated good performance in various conditions and for different time steps (Perrin et al., 2003; Mouelhi et al., 2006; Lobligeois et al., 2014; Ficchì et al., 2016; Santos et al., 2018; Riboust et al., 2019; De Lavenne et al., 2019).

Let us consider a 2D spatial domain (basin, catchment and watershed) $\Omega$ covered by the rectangular grid. For each cell (pixel), the model involves the production and transfer reser-

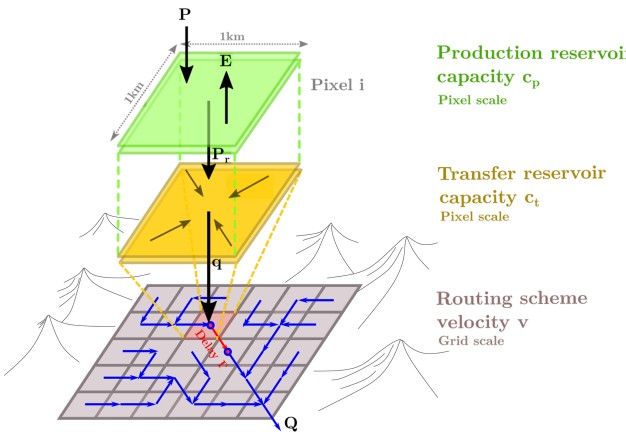

**Figure 1.** General outlines of the GRD model: $P$ represents the local rainfall over one cell; $E$ is the potential evapotranspiration; $P_r$ is the effective rainfall; $q$ is the elementary discharge; and $Q$ is the total routed discharge.

voirs, characterized by capacities $c_p$ and $c_t$, correspondingly, and the discharge generated within each cell is routed between cells with local routing velocity $v$ (see Fig. 1). It means that only three parameters need to be defined for each cell. The integral flow, without distinction between surface, subsurface or groundwater flow, is simulated by the production and transfer reservoirs at the cell level and by the routing scheme at the inter-cell level. Obviously, such model conceptually describes the overall hydrological process rather than its physically meaningful components.

### 2.1.1 The water balance operator

Let $P(t)$ be the local total rainfall (function of time $t$) and $E(t)$ be the local total potential evapotranspiration. For each cell (pixel), a water balance function determines the effective rainfall $P_r$, i.e., the amount of rainfall that will participate to the flow. This function is defined via the following steps. First, the net rainfall $P_n$ and the net potential evapotranspiration $E_n$ are defined from the following equation:

$$\begin{cases} P_n = P - E, \ E_n = 0, & \text{if } P \geq E, \\ P_n = 0, \ E_n = E - P, & \text{if } P < E. \end{cases} \quad (1)$$

Next, the production reservoir is filled by $P_p$, a part of $P_n$. Similarly, the production reservoir is emptied by $E_p$, a part of $E_n$ representing the actual evapotranspiration. The variation of the reservoir level $h_p$ is driven by the following differential equation (Edijatno, 1991):

$$dh_p = \left[ 1 - \left( \frac{h_p}{c_p} \right)^2 \right] dP_n - \frac{h_p}{c_p} \left( 2 - \frac{h_p}{c_p} \right) dE_n. \quad (2)$$

Assuming a stepwise approximation of input variables $P(t)$ and $E(t)$, Eq. (2) is integrated over one time step $\Delta t$ to obtain the amount $P_p$ filling the reservoir and the amount $E_p$

evaporated from it:

$$P_p = c_p \left( 1 - \left( \frac{h_p}{c_p} \right)^2 \right) \frac{\tanh\left( \frac{P_n}{c_p} \right)}{1 + \left( \frac{h_p}{c_p} \right) \tanh\left( \frac{P_n}{c_p} \right)}, \quad (3)$$

$$E_p = h_p \left( 2 - \frac{h_p}{c_p} \right) \frac{\tanh\left( \frac{E_n}{c_p} \right)}{1 + \left( 1 - \frac{h_p}{c_p} \right) \tanh\left( \frac{E_n}{c_p} \right)}. \quad (4)$$

It should be noted that with this discretized temporal formulation, $h_p$ is the reservoir level at the beginning of $\Delta t$ and $P_p$ and $E_p$ are the volume of water gained or lost by the reservoir over $\Delta t$. At the end of $\Delta t$, $h_p$ is updated by adding $P_p$ or removing $E_p$ before progressing to the next time step. Finally, $P_r$ is the part of the net rainfall that does not enter into the production reservoir:

$$P_r = P_n - P_p. \quad (5)$$

One can see that the state of the production reservoir $h_p$ plays the role of the humidity state of the soil. An empty reservoir ($h_p = 0$) means that the soil is completely dry: effective rainfall and evapotranspiration are equal to zero, and all amount of the net rainfall enters into the production reservoir ($E_p = 0$, $P_p = P_n$ and $P_r = 0$). On the contrary, a full reservoir ($h_p = c_p$) means that the soil is completely saturated: no more rainfall enters into the production reservoir, the evapotranspiration is maximal and all the net rainfall produces the effective rainfall ($E_p = E_n$, $P_p = 0$ and $P_r = P_n$).

### 2.1.2 The transfer operator

The effective rainfall $P_r(t)$ fills the transfer reservoir characterized by state $h_t$ and capacity $c_t$. The emission from the transfer reservoir during $\Delta t$ gives the elementary discharge $q$. This transformation is modeled by a conservative operator which is derived from the differential equation describing the evolution of $h_t$ under the mass conservation condition:TS1

$$\frac{dh_t}{dt} + c_t h_t^a = P_r. \quad (6)$$

It has been noticed (Perrin et al., 2003) that Eq. (6) correctly replicates the flooding and drying processes for $a = 5$TS2. This is an empirical knowledge which has no physical justification. Assuming $P_r$ is the impulse function, Eq. (6) is integrated over one time step $\Delta t$ to obtain the expression for $q$:TS3

$$q = h_t - (h_t^{-4} + c_t^{-4})^{-0.25}. \quad (7)$$

More details about the production and the transfer reservoirs can be found in Perrin et al. (2003).

### 2.1.3 The routing scheme (cell to cell)

The total discharge ($Q$) in a cell is then obtained by routing through the catchment all the upstream elementary discharges ($q$). This routing is built on top of a digital elevation model which, for a given node, defines the flow direction. The routing nodes are placed at the center of the corresponding cells.

For the sake of simplicity we describe the routing model in the 1D setting. The total discharge from node $i-1$ to node $i$ is delayed by time

$$\tau(v_{i-1,i}) = \frac{d_{i-1,i}}{v_{i-1,i}}, \tag{8}$$

where $d_{i-1,i}$ and $v_{i-1,i}$ are the distance and the routing velocity between the nodes, respectively. In the simplest implementation, the output discharge (more precisely, the mass over the time step $\Delta t$) is given as

$$Q_i(t) = q_i(t) + Q_{i-1}(t - \tau(v_{i-1,i})), \tag{9}$$

where $Q_i$ is the total discharge in cell $i$, $Q_{i-1}$ is the total discharge in the neighboring upstream cell and $q_i$ is the elementary discharge emitted from the transfer reservoir over time period $\delta t$ at cell $i$. Note that in a 2D case, the second term of the right-hand side of Eq. (9) could be a sum of a few contributions from direct neighboring cells, with their own values of $d$ and $v$. Since no explicit expression for $Q$ is provided, $Q$ is not differentiable with respect to $v$. That is why the above formulation is not suitable for the variational approach, which requires the gradient of the cost function to be computed. In order to tackle this issue we represent the second term in Eq. (9) in the integral form as follows: TS4

$$Q_{i-1}(t - \tau(v_{i-1,i})) =$$
$$\int_{t'=-\infty}^{t} Q_{i-1}(t') \, \delta(t' - \tau(v_{i-1,i})) \, \mathrm{d}t'. \tag{10}$$

Next, instead of the $\delta$ function we use the unscaled Gaussian function, i.e., TS5

$$Q_{i-1}(t - \tau(v_{i-1,i})) =$$
$$\int_{t'=-\infty}^{t} Q_{i-1}(t') \, \omega\left(t' - \frac{d_{i-1,i}}{v_{i-1,i}}, \sigma\right) \mathrm{d}t', \tag{11}$$

where

$$\omega(t, \sigma) = \exp\left(-\frac{t^2}{2\sigma^2}\right). \tag{12}$$

It is easy to see that function Eq. (11) explicitly depends on $v_i$ via $\omega$; therefore the gradient of $Q_{i-1}$ with respect to $v_i$ can be computed. Assuming $Q(t)$ is a constant during a time step

period $\Delta t$, Eq. (11) can be written in the discrete form as follows:

$$Q_{i-1}\left(t - \tau(v_{i-1,i})\right) = \sum_{l=1}^{L} \overline{\beta}_{i,l} \, Q_{i-1}(t - (l-1)\Delta t), \tag{13}$$

where TS6

$$\overline{\beta}_{i,l} = \beta_{i,l} / \sum_{l=1}^{L} \beta_{i,l},$$

$$\beta_{i,l} = w\left(t - (l-1)\Delta t - \frac{d_{i-1,i}}{v_{i-1,i}}, \sigma\right), \; l = 1, \ldots, L$$

and $L$ defines the finite time period (in terms of $\Delta t$) instead of the semi-infinite period considered in Eq. (10). For the given estimate of routing velocities $v_{i-1,i}$, the coefficient $\beta_{i,l}$ does not change with time and, therefore, can be precomputed and saved in memory. The spread coefficient $\sigma$ models the diffusion. The value of parameter $\sigma$ should not be lower than 0.5 to avoid numerical instability (i.e., when a small variation of the routing velocity results in a large variation of "delay"). Therefore, $\sigma = 0.5$ is used in computations. In terms of using the exponential weights the presented routing model resembles the lag-and-route (LR) model described in Laganier et al. (2014) and Tramblay et al. (2010). However, the Gaussian function represents the hydraulic response function in a more realistic way. Indeed, in the lag-and-route method, the kernel function $\omega$ is discontinuous, being zero for $t' > t - \tau$. It means the outflow from cell $i-1$ arrives to cell $i$ in a "shock" manner. If the Gaussian function is used, there is no discontinuity; i.e., outflow from cell $i-1$ arrives to cell $i$ progressively. This scheme is more suitable for cell-to-cell implementation, as it is more stable for direct modeling and the absence of discontinuity is a necessary condition to achieve the differentiability of the forward operator.

### 2.2 Variational-calibration algorithm

Calibrating a distributed model is often difficult due to a number of reasons. First, the total number of the sought parameters can be quite large (high dimensionality). This strictly limits the choice of suitable inference methodologies. Second, there is an identifiability issue given the sparsity of observations in space, the information content of the test signal (rainfall variability) and, possibly, the chosen model structure. The first two can be partially compensated by increasing the observation period or observation frequency to better analyze the system dynamics.

For distributed models the variational-estimation algorithm could be a natural choice, since it is scalable, i.e., it works efficiently for a practically unlimited size of the control vector. That is why this method (branded as 4D-Var) is commonly used in meteorology and oceanography for operational forecasting and reanalysis (Ledimet and Talagrand, 1986; Rabier and Courtier, 1992). The method provides the

exact mode of the posterior distribution by minimizing the cost function defined over the given observation window. The key element of the method is the adjoint model, which provides the precise gradient of the cost function with respect to all elements of the control vector in a single run (Errico, 1997). This allows for the fast-converging gradient-based minimization methods to be used, such as the BFGS (Broyden–Fletcher–Goldfarb–Shanno) or the Newton type. Quite often, the need for development of the adjoint model becomes an obstacle for the practical implementation of this method. Heuristic methods such as the Nelder–Mead algorithm do not require the gradient to evaluate the descent directions but converge slowly and are not suitable for solving problems in high dimensions. The same is true as for the general-purpose statistical methods such as the Markov chain Monte Carlo (e.g., Metropolis–Hastings algorithm), as well as for the methods specially designed for hydrology applications, such as SUFI-2 (Sequential Uncertainty Fitting) (Abbaspour et al., 2007).

Let us consider the rainfall and potential-evapotranspiration fields $P(x,t)$ and $E(x,t)$, $\forall x \in \Omega$. We represent the hydrological model in Sect. 2.1 as an operator $A$ mapping the input fields $\boldsymbol{P}(x,t)$ and $\boldsymbol{E}(x,t)$ into the discharge $Q_k(t)$ at the gauged nodes $x_k \in \Omega$, $k = 1, N_g$:

$$Q_k(t) = A(\boldsymbol{P}(x,t'), \boldsymbol{E}(x,t'), \boldsymbol{h}(x,0), \boldsymbol{\theta}(x), t),$$
$$\forall x \in \Omega, \ t' \in (0,t), \tag{14}$$

where $\boldsymbol{h}(x,t) = (\boldsymbol{h}_p(x,t), \boldsymbol{h}_t(x,t))^T$ is the state vector, which includes the states of production and transfer reservoirs for all cells at time $t$; $\boldsymbol{\theta}(x) = (c_p(x), c_t(x), v(x))^T$ is the parameter vector, which includes the corresponding capacities and the routing velocities at all routing nodes; and $N_g$ is the number of gauged routing nodes, i.e., where discharge observations are available. If the observation period is much longer than the characteristic time of the system (which is the case for calibration or reanalysis), one can use the trivial initial state $\boldsymbol{h}(x,0) = 0$, but consider the observation window $t \in (t^*, T)$, where $t^*$ is the relaxation period. Given the observed inputs $\boldsymbol{P}^*(x,t')$, $\boldsymbol{E}^*(x,t')$, $t' \in (0,t)$ and the output $Q_k^*(t)$, the calibration cost function can be defined as follows:

$$J(\theta) = \sum_{k=1}^{N_g} a_k^{-1}(t^*) \int_{t=t^*}^{T} \left( A(\boldsymbol{P}^*, \boldsymbol{E}^*, 0, \boldsymbol{\theta}, t) - Q_k^*(t) \right)^2 \mathrm{d}t$$
$$+ \alpha \| \mathbf{B}^{-1/2}(\boldsymbol{\theta} - \boldsymbol{\theta}^*) \|_{L^2}^2, \tag{15}$$

where $\mathbf{B}$ is the background error covariance; $\boldsymbol{\theta}^*$ is a prior guess on $\boldsymbol{\theta}$, which comes from special measurements, land expertise or a modeling; $\alpha$ is the regularization parameter; and $a_k$ are the scaling constraints. If we consider

$$a_k(t^*) = \int_{t=t^*}^{T} \left( \langle Q_k^* \rangle - Q_k^*(t) \right)^2 \mathrm{d}t,$$

where $\langle Q_k^* \rangle$ is the temporal mean of $Q_k^*(t)$, then for each $k$ the misfit term becomes $1 - \mathrm{NSE}$, where NSE stands for the Nash–Sutcliffe efficiency criterion (Nash and Sutcliffe, 1970) widely used in hydrology. In essence, Eq. (15) is more or less a standard quadratic cost function similar to the one used in variational data assimilation (4D-Var), where the weight $\alpha$ is introduced to mitigate the uncertainty in $\boldsymbol{\theta}^*$ and $\mathbf{B}$. Let us note that this paper is focused on the parameter calibration problem involving long time series of observations.

We use additional constraints in the form

$$\boldsymbol{\theta}_{\min} \le \boldsymbol{\theta} \le \boldsymbol{\theta}_{\max}, \tag{16}$$

where $\boldsymbol{\theta}_{\min}$ and $\boldsymbol{\theta}_{\max}$ are the bounds which come from the empirical knowledge or physical considerations. Thus, the optimal estimate of the parameters $\hat{\boldsymbol{\theta}}$ is obtained from the condition

$$\hat{\boldsymbol{\theta}} = \arg\min_{\theta} J(\boldsymbol{\theta}), \tag{17}$$

given constraints of Eqs. (14) and (16).

Matrix $\mathbf{B}$ can be represented in the form $\mathbf{B} = \boldsymbol{\sigma}_\theta \cdot \mathbf{I} \mathbf{C} \, \boldsymbol{\sigma}_\theta \cdot \mathbf{I}$, where $\boldsymbol{\sigma}_\theta$ is the vector of mean deviations of $\theta$, $\mathbf{C}$ is the correlation matrix, $\mathbf{I}$ is the identity matrix and "·" stands for the element-wise (Hadamard) product. Next, the scaling of parameters is introduced such that $\boldsymbol{\theta} = \boldsymbol{\theta}_{\min} + \tilde{\boldsymbol{\theta}}(\boldsymbol{\theta}_{\max} - \boldsymbol{\theta}_{\min})$ to ensure that $0 \le \tilde{\boldsymbol{\theta}} \le 1$. Then, the penalty term in Eq. (15) takes the form

$$\alpha \| (\boldsymbol{\theta}_{\max} - \boldsymbol{\theta}_{\min}) \cdot \boldsymbol{\sigma}_\theta^{-1} \cdot \mathbf{I} \mathbf{C}^{-1/2} (\tilde{\boldsymbol{\theta}} - \tilde{\boldsymbol{\theta}}^*) \|_{L^2}^2.$$

Assuming that $(\boldsymbol{\theta}_{\max} - \boldsymbol{\theta}_{\min}) \cdot \boldsymbol{\sigma}_\theta^{-1} = \mathrm{const}$, the cost function of Eq. (15) reads as follows:

$$J(\tilde{\boldsymbol{\theta}}) = \sum_{k=1}^{N_g} a_k^{-1}(t^*) \int_{t=t^*}^{T} \left( A(\boldsymbol{P}^*, \boldsymbol{E}^*, 0, \boldsymbol{\theta}) - Q_k^* \right)^2 \mathrm{d}t$$
$$+ \alpha \| \mathbf{C}^{-1/2}(\tilde{\boldsymbol{\theta}} - \tilde{\boldsymbol{\theta}}^*) \|_{L^2}^2, \tag{18}$$

given

$$\boldsymbol{\theta} = \boldsymbol{\theta}_{\min} + \tilde{\boldsymbol{\theta}}(\boldsymbol{\theta}_{\max} - \boldsymbol{\theta}_{\min}), \ 0 \le \tilde{\boldsymbol{\theta}} \le 1. \tag{19}$$

The results presented in this paper correspond to the simplest approach to regularization: we assume that $\mathbf{C} = \mathbf{I}$, and the regularization parameter is chosen a priori as a small value ($\alpha = 10^{-4}$) to ensure the formal well-posedness of the calibration problem. More sophisticated approaches for regularization (nontrivial correlation matrix $\mathbf{C}$ and the optimal choice of $\alpha$ using the $L$-curve approach) have been utilized (Jay-Allemand et al., 2018) but are not presented in this paper for the sake of simplicity. In practice, the simplifications mentioned above lead to significantly more oscillating parameter fields, which does not seem to have a critical influence on the predictive performance of the model (in the open-loop forecasting, at least).

https://doi.org/10.5194/hess-24-1-2020

Minimization of Eq. (18) given constraints of Eq. (19) is performed by LBFGS-B (limited-memory Broyden–Fletcher–Goldfarb–Shanno bound-constrained; Zhu et al., 1994). The minimization process can be written in the form

$$\tilde{\boldsymbol{\theta}}_{i+1} = \tilde{\boldsymbol{\theta}}_i + \gamma \mathbf{H}^{-1}(\boldsymbol{\theta}_i) R[\boldsymbol{J}'_{\tilde{\boldsymbol{\theta}}}(\boldsymbol{\theta}_i)],$$

$$i = 0, 1, \ldots, \quad \tilde{\boldsymbol{\theta}}_0 = \tilde{\boldsymbol{\theta}}^*, \tag{20}$$

where $\boldsymbol{J}'(\boldsymbol{\theta}_i)$ and $\mathbf{H}^{-1}(\boldsymbol{\theta}_i)$ are the gradient (with respect to $\tilde{\boldsymbol{\theta}}$) and the limited-memory inverse Hessian of Eq. (18) at point $\boldsymbol{\theta}_i$, respectively, $\gamma$ is a descent step, $i$ is the iteration number, and $\boldsymbol{R}$ is the gradient projection operator to account for the box constraints. Let us note that $\mathbf{H}^{-1}(\boldsymbol{\theta}_i)$ is directly computed inside the minimization algorithm in such a way that its norm is always bounded. This serves as an additional regularization; thus the solution $\hat{\theta}$ is always bounded, even for $\alpha = 0$ in Eq. (18), i.e., even without the penalty term. The gradient $\boldsymbol{J}'(\boldsymbol{\theta}_i)$ is obtained by solving the adjoint model. This model has been generated by the automatic differentiation engine Tapenade (Hascoet and Pascual, 2013), then manually optimized and, finally, verified using the standard gradient test.

The background value $\boldsymbol{\theta}^*$ is used both as a starting point for iterations and in the penalty term. Given the fact that the information content of the test signal (rainfall) and observations (discharge) may not be sufficient to uniquely identify the distributed coefficients, evaluating an appropriate $\boldsymbol{\theta}^*$ becomes an important issue. Thus, the overall calibration process involves two steps. In the first step we consider a uniform approximation: $\boldsymbol{c}_p(\boldsymbol{x}) = \overline{\boldsymbol{c}}_p$, $\boldsymbol{c}_t(\boldsymbol{x}) = \overline{\boldsymbol{c}}_t$ and $\boldsymbol{v}(\boldsymbol{x}) = \overline{\boldsymbol{v}}$, $\forall x \in \Omega$. In this case (referred as "uniform calibration") the sought vector $\overline{\boldsymbol{\theta}} = (\overline{\boldsymbol{c}}_p, \overline{\boldsymbol{c}}_t, \overline{\boldsymbol{v}})^{\mathrm{T}}$ consists of just three elements. For such a low-dimensional problem, obtaining the globally optimal estimate $\hat{\overline{\boldsymbol{\theta}}}$ (i.e., the one which corresponds to the global minimum of Eq. 18) is feasible by a variety of methods. In particular, we use a simple global-minimization algorithm, the steepest descent method summarized in Edijatno (1991). In the second step we estimate the distributed parameters using the uniform estimate as a background, i.e., $\boldsymbol{\theta}^* = \hat{\overline{\boldsymbol{\theta}}}$. Here, three unknown parameters for each cell are estimated using the variational algorithm described above. This two-step algorithm is referred as "distributed calibration".

Parameter bounds are defined for each step. Numerical and physical considerations enforce the lower bounds so that $(\boldsymbol{c}_p, \boldsymbol{c}_t, \boldsymbol{v}) > 0$. For the uniform calibration, the upper bounds are chosen to preserve the model dynamics. For example, $5\,\mathrm{m\,s}^{-1}$ is used as the velocity upper bound, since above this value the flow delay does not decrease significantly. For the production and transfer reservoirs the upper bounds are set to 5000 and 2000 mm, respectively, since the higher values do not noticeably change the model dynamics (reservoir states remain almost constant in time). For the distributed calibration, bounds are recomputed as $\boldsymbol{\theta}_{\max} = b\hat{\overline{\boldsymbol{\theta}}}$ and $\boldsymbol{\theta}_{\min} = \hat{\overline{\boldsymbol{\theta}}}/b$ TS7. In particular, $b = 4$ TS8 is used for re-

sults presented in this paper. Further increasing $b$ TS9 does not seem to have any influence on the estimates.

## 2.3 Study area and data

A French watershed, the Gardon d'Anduze, has been considered for testing our model and calibration algorithm. Located in the western Mediterranean region, this catchment and its surrounding have been deeply studied in the framework of the HyMeX program (Drobinski et al., 2014) to understand the processes leading to flash floods. For instance, some studies exploited a number of very detailed field measurements during severe storm events (Braud et al., 2014; Vannier et al., 2014). Others tested hypotheses using physically distributed models, for instance the MARINE model (Roux et al., 2011; Garambois et al., 2013, 2015; Douinot et al., 2016, 2018) or the CéVeNnes (VCN) model (Braud et al., 2010; Vannier et al., 2016). A few conceptual distributed models were also considered in this area, such as those implemented into the ATHYS (Atelier Hydrologique Spatialisé) platform (Bouvier and DelClaux, 1996; Laganier et al., 2014; Tramblay et al., 2010).

The main properties of the Gardon d'Anduze are described in Darras (2015). In brief, this is a steep mountainous watershed with a dense hydrographic network spreading over $540\,\mathrm{km}^2$ in the eastern part of the Cévennes mountains (France). The difference in levels between the highest elevation point and Anduze is about 800 m, and the slope reaches 50 % in the upstream part. A metamorphic (schist) but fractured geological formation dominates the watershed. Water infiltrates very quickly (the saturated hydraulic conductivity is greater than $100\,\mathrm{mm\,h}^{-1}$). The water circulation appends mainly underground, but with very short response times (less than 12 h). This area is governed by a transitional Mediterranean–oceanic climate with warm and dry summers, alleviated by the oceanic influence, followed by recurrent short, intense but persistent heavy rainfalls in autumn and winter, known as *épisode méditerranéen*, generating flash floods.

This watershed is well gauged: at least five stations with continuous data collection are operational here (see Fig. 2 and Table 1). For numerical experiments, the discharge data have been extracted from the HYDRO database of the French ministry in charge of the environment. Rainfall gridded data have been provided by Météo France. It is a pseudo-real-time reanalysis (ANTILOPE – ANalyse par spaTIaLisation hOraire des PrEcipitations – J+1, i.e., available 1 d after the current date), merging radar rainfall estimation with in situ rain gauges. The gridded potential evapotranspiration is computed using interannual air temperature values and the formula developed by Oudin et al. (2005). The temperature data were provided by the SAFRAN reanalysis (Vidal et al., 2010). All the time series have been processed at an hourly time step over the continuous 2007–2018 period. All the gridded data are defined at a $1\,\mathrm{km} \times 1\,\mathrm{km}$ spatial resolution,

**Table 1.** Characteristics of the five gauging stations on the Gardon watershed. QIX2 and QIX10 stand for the quantile discharge, respectively, for 2- and 10-year return periods.

| Rivers and station names | Codes | Surfaces (km$^2$) | QIX2 (m$^3$ s$^{-1}$) | QIX10 (m$^3$ s$^{-1}$) |
|---|---|---|---|---|
| The Gardon de Mialet at Mialet | V7124015 | 219 | 243 | 486 |
| The Gardon de Mialet at Générargues | V7124010 | 244 | 346 | 618 |
| The Gardon de Saint-Jean at Saint-Jean-du-Gard | V7135017 | 158 | 230 | 460 |
| The Gardon de Saint-Jean at Saint-Jean de Corbès | V7135010 | 262 | 320 | 598 |
| The Gardon d'Anduze at Anduze | V7144010 | 543 | 634 | 1300 |

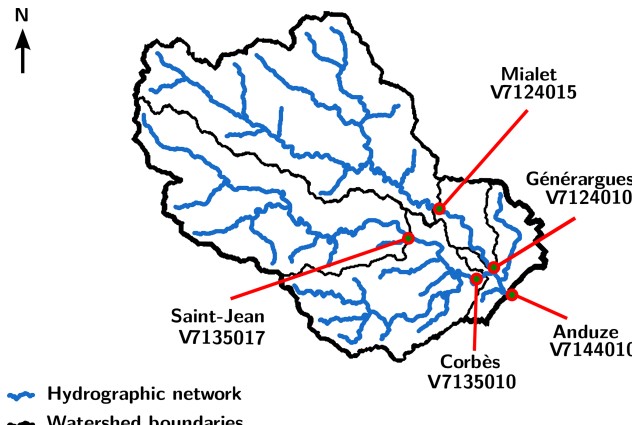

**Figure 2.** The Gardon watershed at Anduze: hydrographic network (blue) and gauging stations V7124015, V7124010, V7135017, V7135010 and V7144010 (red).

over the same 46 km × 40 km domain overlapping the watershed. Furthermore, the flow direction (eight directions) map and the flow accumulation map have been carefully checked, in order to ensure that every cell is connected to the correct downstream cell.

### 2.4 Investigating methodology

The variational algorithm described in Sect. 2.2 is applied to the hydrological model presented in Sect. 2.1, using the Gardon d'Anduze watershed as a benchmark. The number of "active" cells (i.e., included into the watershed) is 540, so the total number of parameters to be calibrated is $3 \cdot 540$.

The calibrated-model validation step consists of checking the model predictive performance (referred as MPP) using the data not involved in calibration. That is, the full set of observations $Q_k^*(t)$, $k = 1, \ldots, N_g$, $t \in (0, T)$ is divided in two complementary subsets: a calibration subset and validation subset. Since $Q^*$ depends on $k$ (defining the spatial distribution of sensors) and $t$, we distinguish the temporal, spatial and spatiotemporal validation, following the split sample test defined by Klemes (1986). In particular, we divide the whole period in two parts: $P1$ from 1 January 2008 to 1 January 2013 and $P2$ from 1 January 2013 to 1 January 2018. Each period of $P1$ and $P2$ can be considered as a cal-

ibration or validation period. A model warm-up of 1 year is performed before starting the simulations. We assume that 1 year is enough, according to recommendations by Perrin et al. (2003).

If data from a station are used in calibration, the corresponding catchment is called the "calibration catchment"; otherwise it is called the "validation catchment".

Both the calibration quality and the MPP in validation are measured using the NSE criterion and the Kling–Gupta efficiency (KGE) criterion (Gupta et al., 2009).

The spatiotemporal measured discharge data are partitioned into calibration and validation complementary sets. Then, we refer to:

a. "temporal validation" if MPP is evaluated for all calibration catchments over the validation period

b. "spatial validation" for all validation catchments over the calibration period

c. "spatiotemporal validation" for all validation catchments over the validation period.

The following numerical experiments have been performed:

1. *Calibration uniform-5-station and calibration distributed-5-sta*. These are uniform and distributed calibration, respectively, using discharge data from all five gauging stations, with calibration periods $P1$ or $P2$. This is followed by temporal validation; i.e., the parameter estimate obtained using data from $P1$ is validated on data from $P2$, and vice versa. The appends uniform-5-sta and distributed-5-sta to the "temporal validation" in Fig. 3 indicate the relevant calibration procedure.

2. *Calibration uniform-1-sta and calibration distributed-1-sta*. These are uniform and distributed calibration, respectively, using discharge data from one downstream gauging station (Anduze), with calibration periods $P1$ or $P2$. This is followed by spatial validation using data from the remaining four gauging stations (Générargues, Mialet, Saint-Jean and Corbès), with calibration periods $P1$ or $P2$ and by spatiotemporal validation on data from the remaining four gauging stations but for validation periods $P2$ or $P1$.

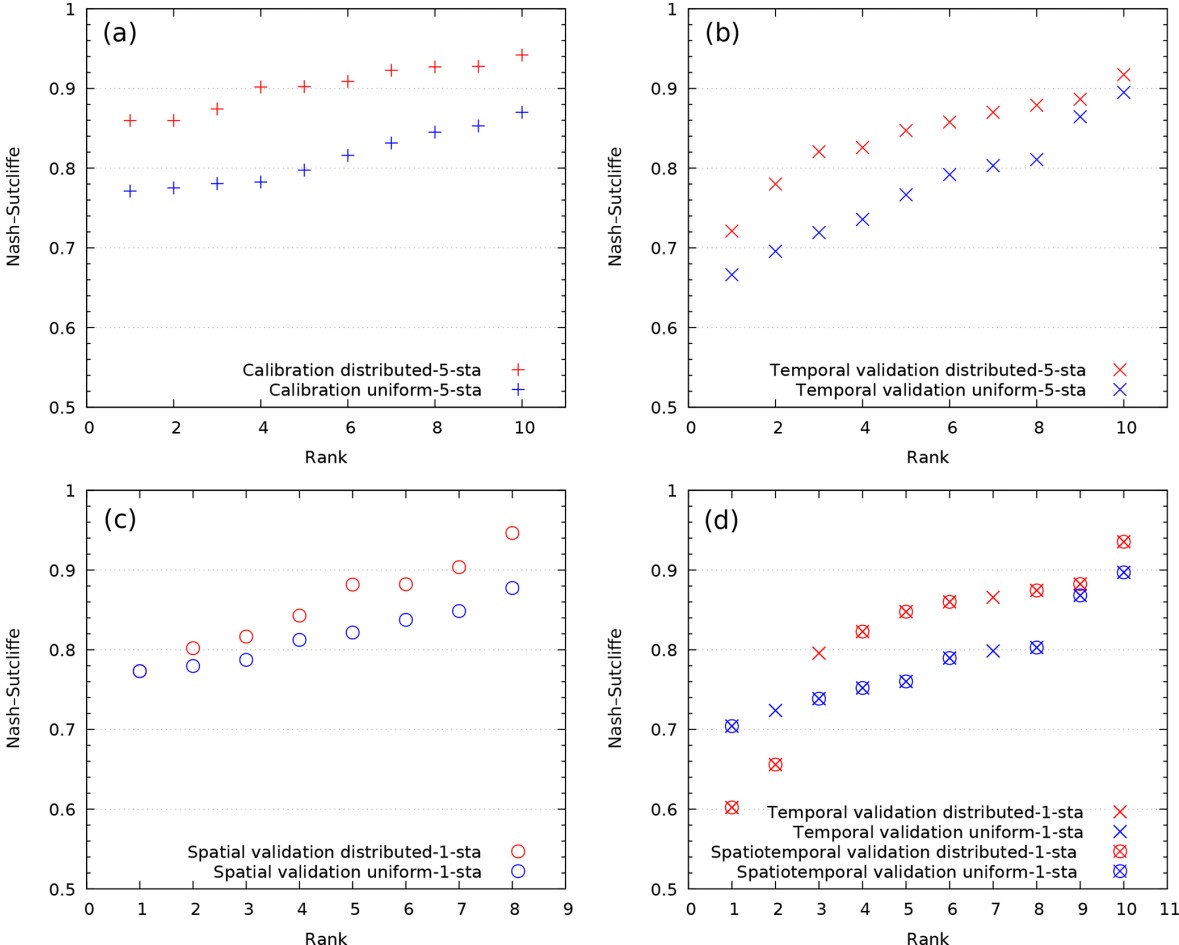

**Figure 3.** "Statistical" distribution of the NSE criterion comparing distributed vs. uniform calibration and the corresponding validation results: calibration of experiment 1 (**a**), temporal validation of experiment 1 (**b**), spatial validation of experiment 2 (**c**) and spatiotemporal validation of experiment 2 (**d**).

3. *Ensemble distributed-1-sta*. This is multiple distributed calibration using discharge data from the Anduze gauging station, starting from different uniform priors.

The main purpose of experiments 1 and 2 is to quantify the anticipated MPP improvement, achieved in "distributed calibration". Given the rainfall data, we compare the ability of the calibrated model to predict the discharge at observation points without using the discharge data for the validation period (experiment 1, temporal validation); then we compare the ability to resolve the spatial distribution of discharge in ungauged areas, given the discharge data for the validation period at the catchment outlet (experiment 2, spatial validation) and the ability to predict the discharge in ungauged areas and without using the discharge data for the validation period (experiment 2, spatiotemporal validation). Clearly, the latter is the ultimate performance test. The purpose of experiment 3 is to investigate the stability of the parameter estimates with respect to their priors. This is necessary to assess

the impacts of equifinality and limitations related to the local-search minimization.

In addition, the current MPP is analyzed for the eight major flood events for periods $P1$ and $P2$, listed in Table 2. Three other criteria from (Artigue et al., 2012) are used, which compare the magnitude and the synchronization of the modeled and observed flood peak.

1. The percentage peak discharge $P_D$:

$$P_D = 100 \times \frac{Q_{max}}{Q_{max}^*}, \tag{21}$$

where $Q_{max} = Q(t_m)$ and $Q_{max}^* = Q^*(t_m^*)$ are the predicted and the observed maximum discharges, respectively. These maximum values are achieved at different time instants $t_m$ and $t_m^*$, within the chosen time period.

2. The synchronous percentage of the peak discharge $S_{PPD}$:

$$S_{PPD} = 100 \times \frac{Q(t_m^*)}{Q_{max}^*}. \tag{22}$$

**Table 2.** Selection of major floods for each period $P1$ and $P2$ at station Anduze (V7144010)

| Event names | Date | Maximum peak discharge ($m^3\,s^{-1}$) |
|---|---|---|
| Event 1/P1 | 21 Oct 2008–23 Oct 2008 | 1020 |
| Event 2/P1 | 31 Oct 2008–7 Nov 2008 | 1030 |
| Event 3/P1 | 31 Jan 2009–5 Feb 2009 | 390 |
| Event 4/P1 | 31 Oct 2011–8 Nov 2011 | 1070 |
| Event 1/P2 | 17 Sep 2014–21 Sep 2014 | 1080 |
| Event 2/P2 | 9 Oct 2014–16 Oct 2014 | 1180 |
| Event 3/P2 | 12 Sep 2015–15 Sep 2015 | 1120 |
| Event 4/P2 | 27 Oct 2015–29 Oct 2015 | 1530 |

3. The peak delay $P_D$:

$$P_D = t_m - t_m^*. \tag{23}$$

# 3 Results

## 3.1 Performance analysis

The results associated with experiments 1 and 2 are presented in Fig. 3 in a "statistical" form (as a distribution of the NSE criterion ranked in increasing order), without relation to the period or gauge station.

Figure 3a shows the results of the "uniform" and "distributed" calibration from experiment 1. All five gauging stations are involved in calibration over two periods; thus we have $5 \cdot 2$ calibration points. One can see that the distributed calibration allows for a much better approximation of the observed discharge than the uniform one. This result confirms that the data assimilation procedure works. It also suggests that hydrological properties of the catchment are not uniform and may be captured using distributed parameters. On the other hand, the mismatch between the predicted and observed discharges remains significant. Since the model looks overparameterized and the regularization parameter in Eq. (18) is very small, this indicates that a few issues, either alone or in combination, may be presented: corrupted data (the data used are not synthetic), a deficient model structure (such as an unaccounted sink term, for example) or a local minimum is reached instead of the global one. Some of these issues will be later discussed. Figure 3b shows the same results for experiment 1, but they are this time in temporal validation. One can see that the distributed calibration allows for achieving a better "global" temporal MPP than the uniform one.

Figure 3c shows the results of spatial validation of experiment 2 (i.e., calibrating only with data from the Anduze gauging station). The remaining four gauging stations are involved in validation for two time periods; thus we have $4 \cdot 2$

spatial validation points. The calibration results for the Anduze station are not presented graphically, but the NSE values achieved are (for $P1$ and $P2$, respectively) uniform of 0.82 and 0.78 and distributed of 0.93 and 0.88, whereas the KGE values are uniform of 0.81 and 0.71 and distributed of 0.89 and 0.69. One can see once again that the distributed calibration allows for clearly achieving a better global spatial MPP than the uniform one. Finally, Fig. 3d shows the results of spatiotemporal validation (experiment 2), evaluated over the period not used for calibration. As before, only data from the Anduze gauging station are used for calibration, but data from all five gauging stations from a different time period are used for validation, giving $5 \cdot 2$ validation points. Two of them (in "×") show the corresponding temporal validation results at the Anduze gauge station; others (shown in "⊗") are the spatiotemporal validation results at the remaining gauge stations. One can see that the spatiotemporal MPP is generally better if the distributed calibration is used.

## 3.2 Period-based analysis

To get a more detailed view of the results of experiment 2, Fig. 4 relates each NSE value to the particular time period and gauging station. Here, Fig. 4a presents the NSE values calculated over the period $P1$, with parameters optimized over the same period (calibration); Fig. 4b presents the NSE values calculated over $P1$, with parameters optimized over $P2$ (validation). Conversely, Fig. 4c contains the NSE values calculated over the period $P2$, with parameters optimized over the same period; Fig. 4d presents the values calculated over $P2$, with parameters optimized over $P1$. All panels present results for the uniform and distributed parameters (blue and red, respectively). The similar figure has been obtained for experiment 1 but not presented here, since the results are also similar. Based on this figure the following observations can be made:

1. Considering the results obtained with uniform parameters (blue), the NSE values are always better (larger) when calculated over the period $P1$, whatever case is considered: calibration (+), spatial validation (o), temporal validation (×) or spatiotemporal validation (⊗).

2. Considering the results obtained with distributed parameters (red), better results are always obtained when the NSE values are calculated over $P1$ in calibration (+) and temporal validation (×). But in spatial validation (o), better results are obtained on $P2$, for two out of four upstream stations: Mialet and Corbès (V7124015 and V7135010, respectively).

3. Comparing the NSE values for the uniform (blue) and the distributed (red) calibration, we notice that for the latter we obtain better results, except in two cases: in spatial validation over $P1$ for Mialet and Corbès (V7124015 and V7135010) and in spatiotemporal validation over $P2$ for Corbès and Saint-Jean (V7135010

and V7135017). Note that in both cases the distributed parameters have been calibrated over $P1$.

To complete the analysis, the MPP obtained in spatiotemporal validation at Mialet (V7124015) and Saint-Jean (V7135017) – two stations identified as "particular" just above – have been studied in more detail, for $2 \cdot 4$ flood events described in Table 2, corresponding to the four biggest floods over each period. The corresponding hydrographs are plotted in Fig. 5, and the resulting MPP criteria of NSE, KGE, PPD, $S_{PPD}$ and $P_D$ are presented in Tables 3 and 4. Based on these results, the following conclusions can be drawn.

1. Flood events at those stations which occur during the period $P1$ are well simulated using both a uniform and distributed set of parameters calibrated over $P2$ ($\overline{\text{NSE}} > 0.70$ and $\overline{\text{KGE}} > 0.60$). In particular, the prediction of the event $3/P1$ is noticeably improved for the distributed calibration. The simulated flood peak is well synchronized with the observed one, though slightly shifted when the distributed calibration is used. Note that these minor shifts are not critical for the peak discharge prediction, as PPD and $S_{PPD}$ remain similar.

2. In contrast, the prediction of flood events which occur during $P2$ is unsatisfactory. Moreover, the results are even worse for distributed calibration. Modeled flood peaks are severely underestimated at station Saint-Jean (V7135017) for all four events, both for uniform and distributed calibration. At station Mialet (V7124015) the peak discharge is overestimated for the distributed calibration.

These results are consistent with those from Fig. 4 in spatiotemporal validation ($\otimes$), for the same two stations. A smaller cost achieved during calibration does not necessary imply a better MPP, indicating an excessive assimilation of errors associated with $P1$.

Figures 6 and 7 represent the maps of the calibrated parameters obtained in experiment 1 and experiment 2, respectively, whereas Table 5 provides the corresponding uniform values $\hat{\bar{\theta}}$ used as priors in the variational-estimation step. Comparing panels a–c and d–f in Fig. 7 corresponding to different time periods $P1$ and $P2$, one can notice a significant difference between the calibrated capacities $c_p$ and $c_t$. The spatial variability of $c_p$ and $c_t$ is also higher when the calibration is performed on period $P1$. One can notice that the results of the uniform calibration are also different for both periods. Concerning the routing velocity $v$, the maps look rather similar. The velocity changes occur mainly at the cells located in the drainage (see Fig. 2), because the sensitivity of the cost function with respect to the velocities in these cells is the largest. The trend is that these velocities increase during calibration. Note that the velocities above $5 \text{ m s}^{-1}$ do not affect significantly the delay between cells. Moreover, the routing velocities obtained are generally consistent with

the fast hydrological response of the Gardon watershed (Darras, 2015). A few questions concerning the results in Figs. 6 and 7 should be answered. For example, why are the estimates associated with periods $P1$ and $P2$ so different? Is it an "algorithmic" issue (for example, due to the use of the local-search minimization, uncorrelated priors, etc.), which can be resolved by improving the calibration algorithm, or a "fundamental" issue related to identifiability (which depends of the data available, properties of the model under investigation, and its adequacy regarding the real natural system and its physical characteristics)? We try to answer this question in the next section.

### 3.3 Stability analysis

An inverse problem is well-posed if a solution to the problem exists, is unique and is continuous with respect to the input data. Let us consider a mathematical model describing a certain physical phenomenon and assume that some variables of this phenomenon are partly observed. If the model is perfect (adequate) and observations are exact then, given the input, the model's output has to match observations. This implies that the minimum of a cost function penalizing the mismatch (mismatch functional) equals zero. Parameter calibration problems are often ill-posed in terms of the uniqueness (equifinality): there may exist a set of solutions for which the mismatch functional equals zero. However, the "true" value of parameters does exist as one element of this set. If the model involved is nonlinear with respect to its parameters, the mismatch functional may contain additional minimum points where its value is not zero. While the corresponding parameters do not belong to the set of solutions, in the local-search minimization methods such points are interpreted as solutions. Adding a penalty (regularization) term to the mismatch functional allows one particular solution, which is the "closest" to the prior in terms of a chosen norm, to be defined. This makes the problem formally well-posed but, in practical terms, transforms the non-uniqueness issue into the issue of choosing the prior. Thus, investigating the stability of the estimates with respect to the priors is an important step in the design and validation of a calibration algorithm. In particular, such an analysis allows for the parts of the solution dominated by observations and by the prior to be distinguished.

A straightforward approach implies solving an ensemble of calibration problems involving random (or quasi-random) uniform priors subjected to box constraints of Eq. (16). Let us remember that the priors are used in two ways: as starting points for minimization and to define the penalty term. The influence of the latter has been neglected by choosing small $\alpha$ in Eq. (18). Let $\hat{\theta}|_{\overline{\theta}_l}$ be the parameter estimate conditioned on the uniform prior $\overline{\theta}_l$, $l = 1, \ldots, L$ from the ensemble of priors of size $L$. Then, the stability for each element of the parameter vector is measured by the standard deviation (SD) given as follows:

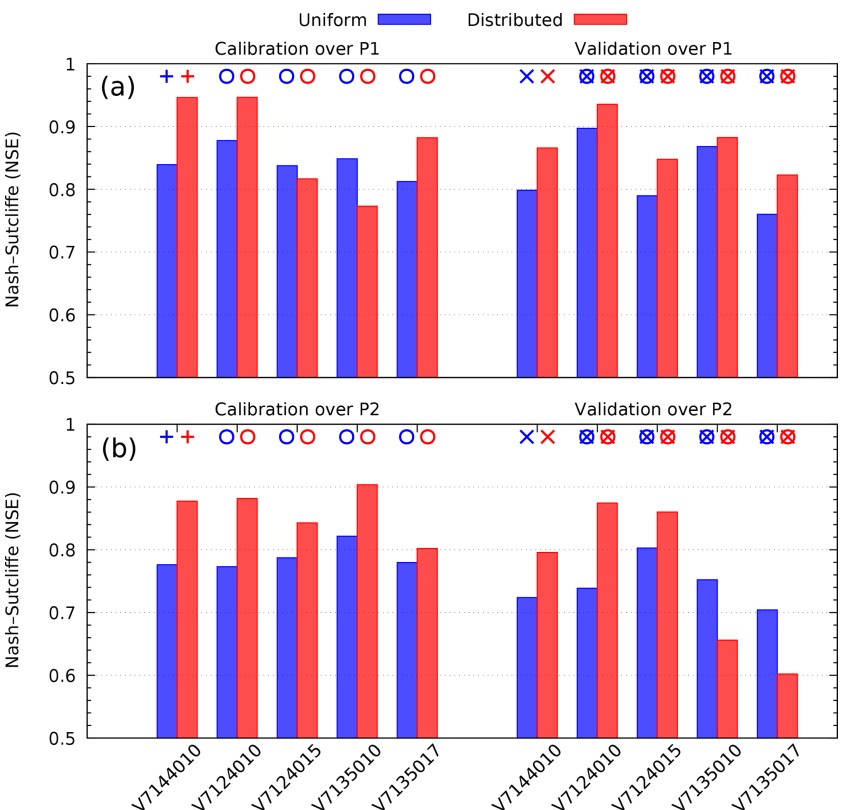

**Figure 4.** NSE criterion calibration and corresponding validation results for experiment 2 (one upstream gauging station), by stations and periods: $P1$ **(a)** and $P2$ **(b)** for uniform parameters (blue) and distributed parameters (red).

**Table 3.** MPP criteria (NSE, KGE, $S_{PPD}$ and $P_D$) computed for major flood events selected over the period $P1$, using distributed (dist) and uniform (unif) calibration over $P2$.

|  |  | NSE |  | KGE |  | PPD |  | $S_{PPD}$ |  | $P_D$ (h) |  |
|---|---|---|---|---|---|---|---|---|---|---|---|
|  |  | dist | unif | dist | unif | dist | unif | dist | unif | dist | unif |
| Event 1/P1 | V7124015 | 0.38 | 0.64 | 0.36 | 0.47 | 47.00 | 54.72 | 40.15 | 54.72 | 1 | 0 |
| Event 1/P1 | V7135017 | 0.71 | 0.64 | 0.84 | 0.59 | 80.69 | 121.23 | 77.15 | 121.23 | 1 | 0 |
| Event 2/P1 | V7124015 | 0.95 | 0.93 | 0.90 | 0.91 | 92.33 | 87.73 | 92.33 | 87.73 | 0 | 0 |
| Event 2/P1 | V7135017 | 0.79 | 0.87 | 0.68 | 0.84 | 66.26 | 79.28 | 64.43 | 79.28 | −1 | 0 |
| Event 3/P1 | V7124015 | 0.92 | 0.68 | 0.91 | 0.57 | 102.38 | 126.21 | 100.43 | 126.21 | −1 | 0 |
| Event 3/P1 | V7135017 | 0.76 | 0.20 | 0.57 | 0.18 | 119.18 | 141.48 | 115.08 | 134.26 | −2 | −3 |
| Event 4/P1 | V7124015 | 0.93 | 0.92 | 0.82 | 0.81 | 94.62 | 120.92 | 94.62 | 120.92 | 0 | 0 |
| Event 4/P1 | V7135017 | 0.71 | 0.86 | 0.47 | 0.63 | 90.10 | 100.70 | 87.82 | 100.70 | 1 | 0 |
| Average |  | 0.77 | 0.72 | 0.69 | 0.63 | 86.57 | 104.03 | 84.00 | 103.13 | −0.13 | −0.38 |

$$\boldsymbol{\sigma}_\theta = \text{sign}(a)\sqrt{|a|},$$

$$a = \frac{1}{L}\sum_{l=1}^{L}\left[\left(\hat{\boldsymbol{\theta}}|_{\overline{\boldsymbol{\theta}}_1} - \langle\hat{\boldsymbol{\theta}}\rangle\right)^2 - \left(\overline{\boldsymbol{\theta}}_1 - \langle\overline{\boldsymbol{\theta}}\rangle\right)^2\right], \quad (24)$$

where $\langle\overline{\boldsymbol{\theta}}\rangle$ and $\langle\hat{\boldsymbol{\theta}}\rangle$ are the ensemble average of priors and estimates, respectively. One can see that the negative values of $\boldsymbol{\sigma}_\theta$ correspond to the case when the solution tends to approach the same value for all priors; i.e., it is dominated (or stabilized) by observations, whereas the positive values of $\boldsymbol{\sigma}_\theta$ are for the case when the solution is dominated by the prior.

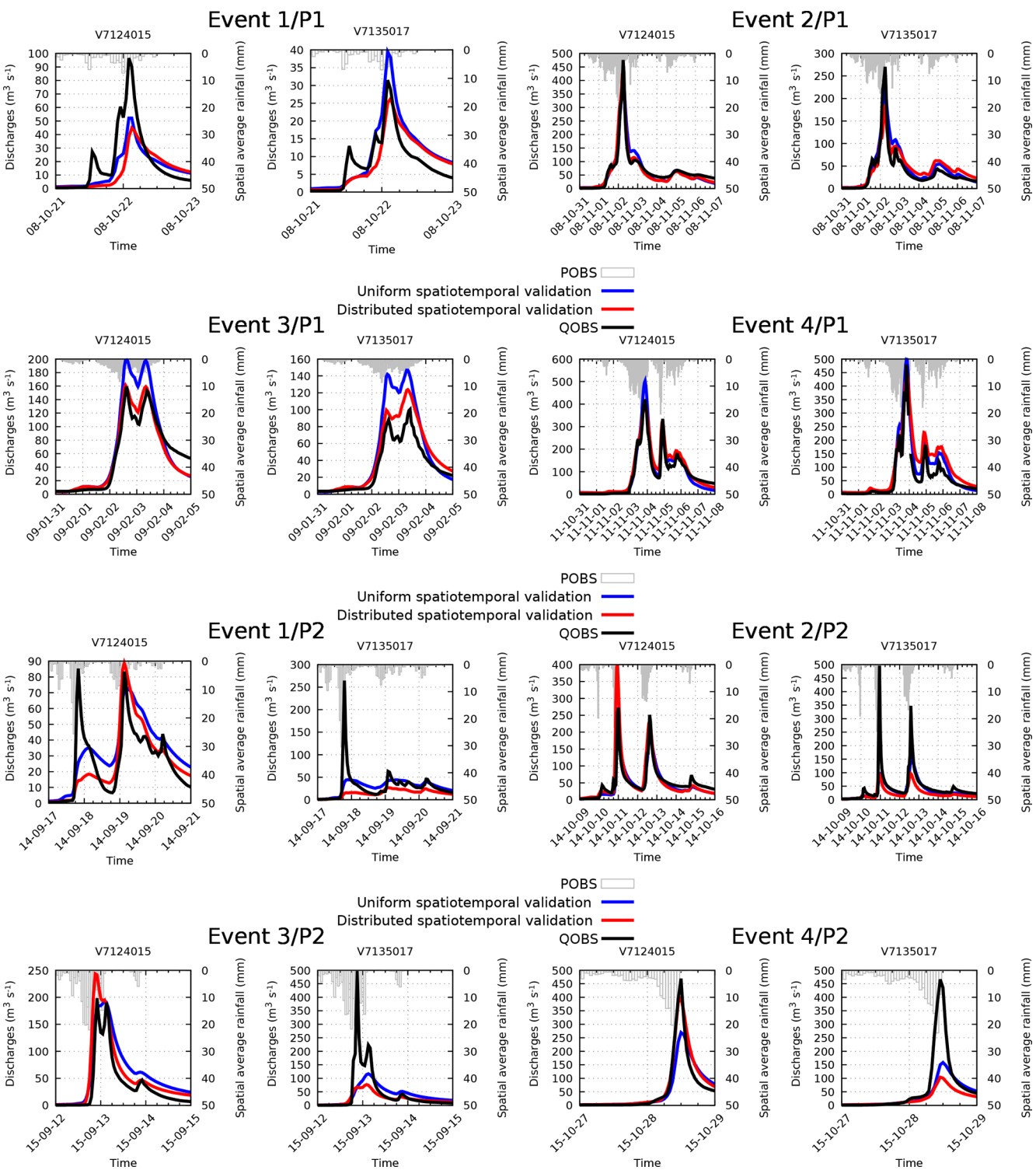

**Figure 5.** Spatiotemporal predicted discharges (with the distributed and uniform set of parameters) and observed discharges during the height major events issued from period $P1$ and $P2$ at stations V7124015 and V7135017. Please note that the date format in this figure is two-digit year-month-day. QOBS: observed $Q$; POBS: observed $P$.

**Table 4.** MPP criteria (NSE, KGE, $S_{PPD}$ and $P_D$) computed for major flood events selected over the period $P2$, using distributed (dist) and uniform (unif) calibration over $P1$.

|  |  | NSE | | KGE | | PPD | | $S_{PPD}$ | | $P_D$ (h) | |
| --- | --- | --- | --- | --- | --- | --- | --- | --- | --- | --- | --- |
|  |  | dist | unif | dist | unif | dist | unif | dist | unif | dist | unif |
| Event 1/P2 | V7124015 | 0.43 | 0.45 | 0.72 | 0.64 | 105.32 | 86.69 | 17.00 | 30.70 | 31 | 32 |
| Event 1/P2 | V7135017 | −0.10 | 0.24 | −0.12 | 0.22 | 9.69 | 16.27 | 5.14 | 14.15 | 31 | 2 |
|  |  |  |  |  |  |  |  |  |  |  |  |
| Event 2/P2 | V7124015 | 0.43 | 0.86 | 0.51 | 0.90 | 148.00 | 73.79 | 126.57 | 73.79 | −1 | 0 |
| Event 2/P2 | V7135017 | 0.31 | 0.47 | 0.13 | 0.37 | 18.76 | 29.48 | 17.86 | 17.51 | 39 | 40 |
|  |  |  |  |  |  |  |  |  |  |  |  |
| Event 3/P2 | V7124015 | 0.59 | 0.63 | 0.41 | 0.34 | 124.36 | 98.42 | 123.30 | 91.35 | −1 | 4 |
| Event 3/P2 | V7135017 | 0.27 | 0.35 | 0.09 | 0.27 | 14.96 | 22.68 | 13.08 | 13.33 | 5 | 6 |
|  |  |  |  |  |  |  |  |  |  |  |  |
| Event 4/P2 | V7124015 | 0.94 | 0.79 | 0.83 | 0.73 | 85.69 | 58.20 | 85.69 | 58.20 | 0 | 0 |
| Event 4/P2 | V7135017 | 0.26 | 0.46 | 0.02 | 0.24 | 21.64 | 32.94 | 21.64 | 31.17 | 0 | 1 |
| Average |  | 0.39 | 0.53 | 0.32 | 0.46 | 66.05 | 52.31 | 51.29 | 41.28 | 13 | 10.625 |

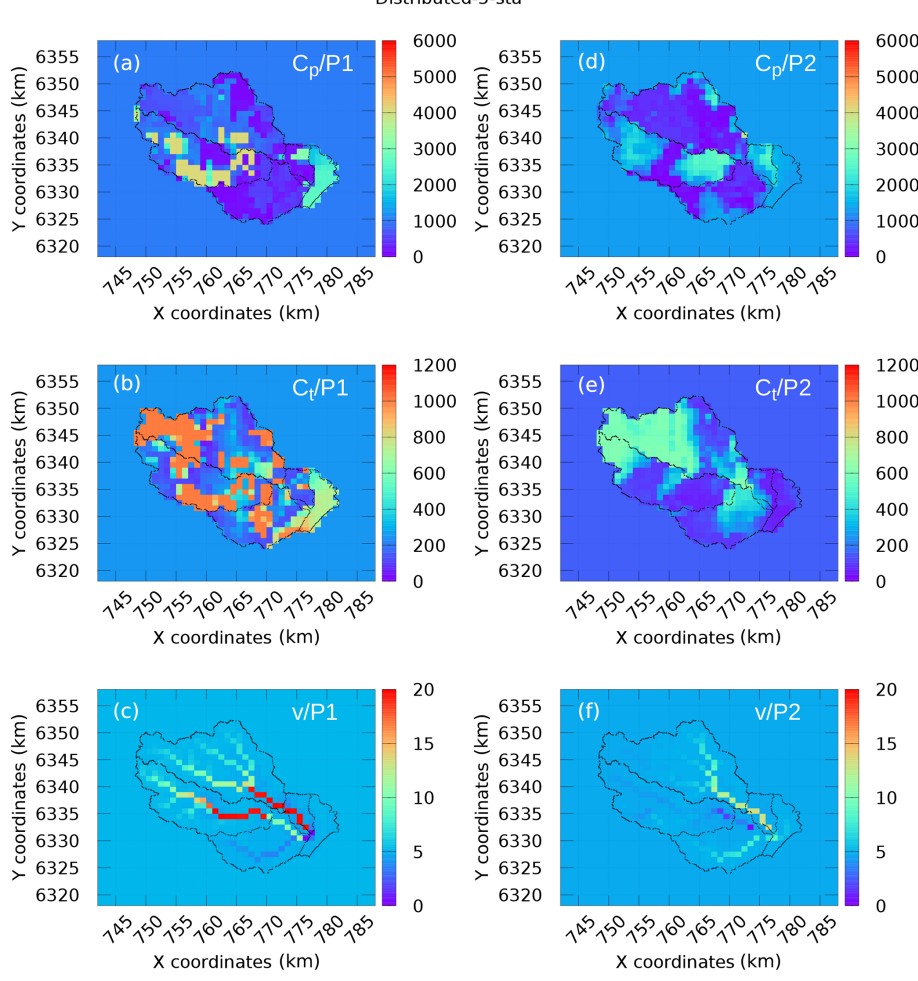

**Figure 6.** Maps of the calibrated coefficients (experiment 1 and 5-sta): **(a–c)** data from $P1$ and **(d–f)** data from $P2$.

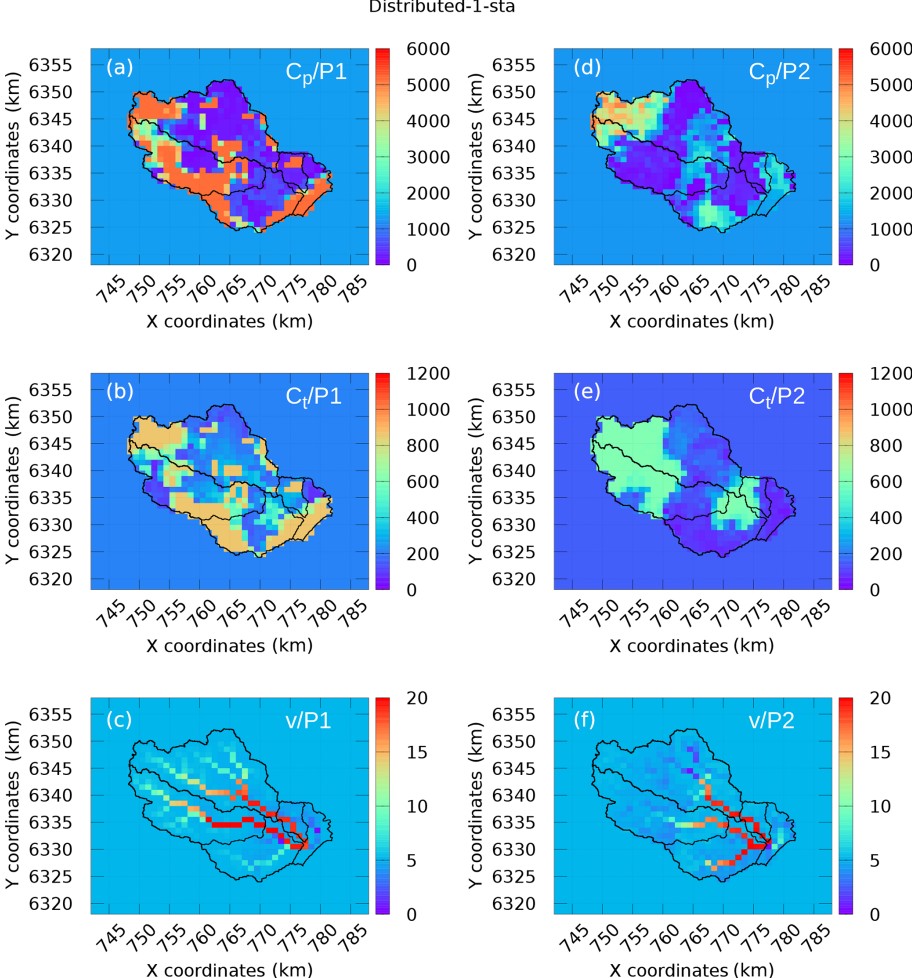

**Figure 7.** Maps of the calibrated coefficients (experiment 3 and 1-sta): **(a–c)** data from $P1$ and **(d–f)** data from $P2$.

**Table 5.** Optimal uniform set of parameters for experiments uniform-5-sta and uniform-1-sta.

| Parameters | Period $P1$ (5-sta/1-sta) | Period $P2$ (5-sta/1-sta) |
|---|---|---|
| $c_p$ (mm) | 1014.2 and 1276.5 | 1288.5 and 1202.3 |
| $c_t$ (mm) | 250 and 214.7 | 150.7 and 150.1 |
| $v$ (m s$^{-1}$) | 5 and 5 | 5 and 5 |

In reality, the conceptual GRD model used in this study is a fairly crude approximation of the hydrological phenomenon. Thus, no solution can be considered as a "truth", but it is rather as an interpretation of data, given the model and the judgment criteria (cost function). Moreover, the measured data (test signal and observations) are not perfect. These imperfections result in a "generalized observation error". The stability analysis described above can also be applied in this case, though understanding of results is more difficult. For example, the ensemble average of the mismatch functional values (achieved in minimization) can be considered as a ref-

erence level. The parameter estimates which correspond to the values around this level can be considered as possible solutions, whereas the outliers must be discarded.

The stability analysis (experiment 3) has been performed for $L = 16$ configurations of uniform priors $\overline{c}_p$ and $\overline{c}_t$ (see Table 6) with the same routing velocity $\overline{v} = 5$ m s$^{-1}$. The table shows the values of the cost function $J(\theta)$ of Eq. (18) for $\theta$ before and after minimization.

One can see that all minimization processes for a chosen assimilation period converge to a similar value of the cost function: for $P1$ the mean value is 0.055, and the SD is 0.002, i.e., about 3.6 % of the mean; for $P2$ the corresponding numbers are 0.129, 0.004 and 3.1 %. No obvious outliers have been observed, which indicates that the cost function surface is sufficiently regular and the issue of "local search vs. global search" is not critical. The latter is not surprising, since, considering the operators involved, the model seems to be mildly nonlinear.

The spatial distributions of $\boldsymbol{\sigma}_\theta$ are presented in Fig. 8, where the stable areas of the estimated parameter fields, i.e.,

**Table 6.** Initial and final values (separated by a forward slash) of the cost function of Eq. (18) for different uniform priors for periods $P1$ **(a)** and $P2$ **(b)** for experiment 3.

| **(a)** | | 1 $\bar{c}_p = 425.5$ | 2 $\bar{c}_p = 851.0$ | 3 $\bar{c}_p = 1276.5$ | 4 $\bar{c}_p = 1702.0$ |
|---|---|---|---|---|---|
| 1 | $\bar{c}_t = 71.5$ | 1.20/0.054 | 0.689/0.054 | 0.528/0.057 | 0.451/0.057 |
| 2 | $\bar{c}_t = 143.1$ | 0.563/0.055 | 0.280/0.056 | 0.210/0.057 | 0.195/0.058 |
| 3 | $\bar{c}_t = 214.7$ | 0.350/0.057 | 0.186/0.053 | 0.161/0.054 | 0.177/0.054 |
| 4 | $\bar{c}_t = 286.2$ | 0.280/0.057 | 0.185/0.055 | 0.185/0.056 | 0.217/0.052 |
| **(b)** | | 1 $\bar{c}_p = 413.0$ | 2 $\bar{c}_p = 825.9$ | 3 $\bar{c}_p = 1238.9$ | 4 $\bar{c}_p = 1651.9$ |
| 1 | $\bar{c}_t = 50.0$ | 1.39/0.130 | 0.924/0.130 | 0.776/0.131 | 0.707/0.130 |
| 2 | $\bar{c}_t = 100.1$ | 0.642/0.125 | 0.369/0.126 | 0.296/0.132 | 0.288/0.125 |
| 3 | $\bar{c}_t = 150.1$ | 0.396/0.134 | 0.252/0.124 | 0.224/0.130 | 0.24/0.129 |
| 4 | $\bar{c}_t = 200.1$ | 0.325/0.131 | 0.26/0.121 | 0.258/0.136 | 0.286/0.126 |

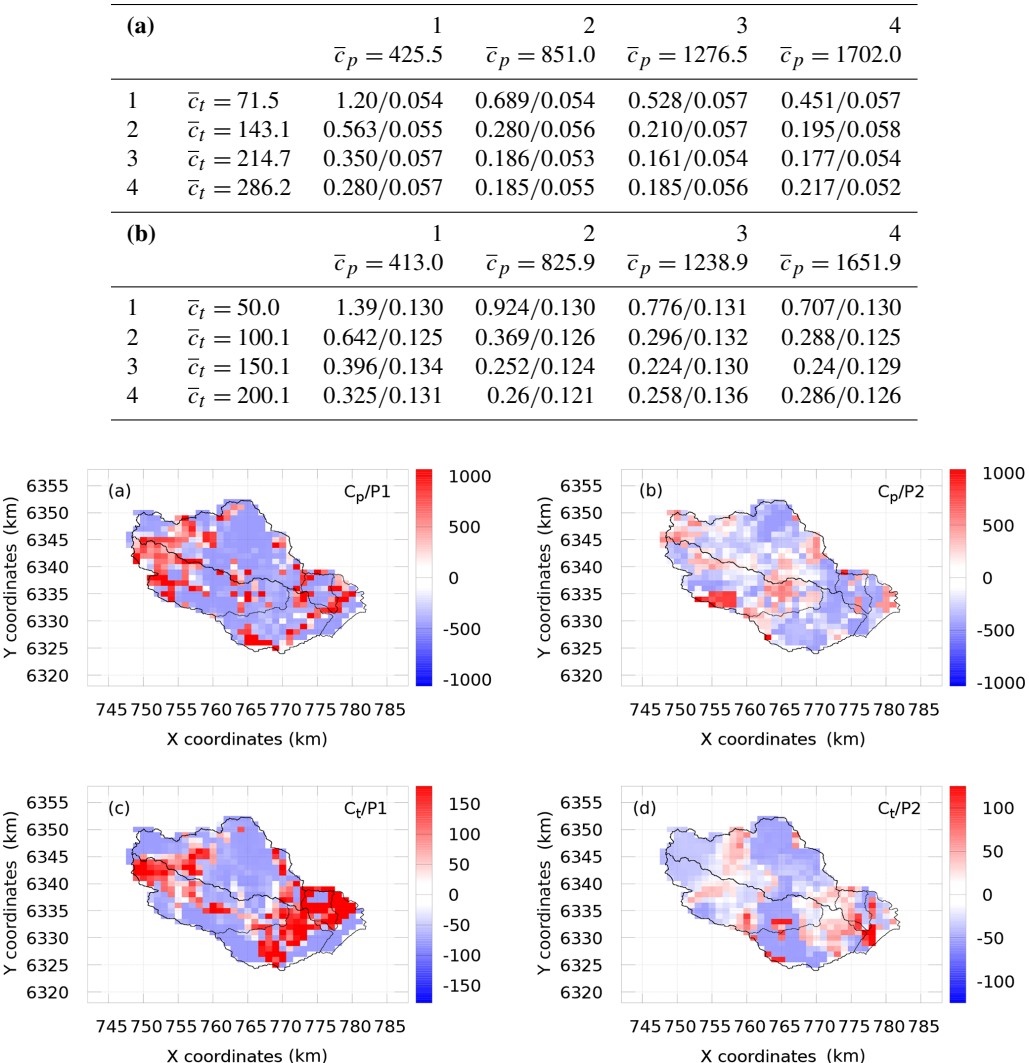

**Figure 8.** Maps of stability measure $\sigma_\theta$ for $c_p$ **(a, b)** and $c_t$ **(c, d)**, for periods $P1$ **(a, c)** and $P2$ **(b, d)**

those influenced by observations, are shown in blue. The purpose of this analysis is to understand why the estimates associated with periods $P1$ and $P2$ are so different. In particular, the question rises concerning the high capacities obtained for $P1$; see red areas in Fig. 7a–c for $c_p$ and dark-yellow areas for $c_t$. In contrast, the capacities obtained for $P2$ are the lowest in the corresponding areas. However, comparing Figs. 7 and 8, we notice that these "questionable" areas in Fig. 7 largely coincide with the blue (stable) areas in Fig. 8. This clearly indicates that the corresponding estimates are determined by the observed data used in each case. Thus, one may suspect here the data information content issue, the data interpretation criteria (cost function) issue and/or the model adequacy issue, rather than algorithmic issues. This is the main conclusion drawn from the stability analysis. One can

also notice that the estimates associated with period 2 ($P2$) are generally more stable than those associated with period 1 ($P1$), whereas the ensemble-average cost function value achieved for $P2$ is larger than for $P1$, i.e., 0.129 compared to 0.055. While a better minimization result has been achieved for $P1$, it looks like more error has been assimilated in this case.

## 4 Discussion and conclusions

The validation results presented in Sects. 3.1 and 3.2 confirm that the distributed calibration globally improves the temporal and spatial MPP (measured by NSE) as compared to the uniform calibration. The conclusions are less definite with

the spatiotemporal MPP, but this is the most difficult performance test. Moreover, the global improvement does not mean that any particular event could be better predicted, as follows from Sect. 3.2. We observe that, on the one hand, the capacity $c_p$ and $c_t$ estimates obtained using different calibration periods are quite different, whereas, on the other hand, for a chosen calibration period these estimates are relatively stable with respect to their priors. The latter means that the difference is rather due to different data involved in calibration than due to the algorithmic or identifiability problems (see Sect. 3.3). Let us note that the cross-validation experiments could help the best set of parameters to be selected. For example, the MPP analysis above suggests that the sets calibrated using data from $P2$ should be preferred.

It seems that the routing velocity $v$ is the most stable among all estimated parameters: the velocity fields plotted in Figs. 6 and 7 remain similar for different experiments. Looking at the hydrographic network in Fig. 2, one can see that the velocities are generally much higher along the main drains than on the side slopes, which is in agreement with the true physical behavior of the system, though the routing scheme applied is conceptual. One can notice a few occasions where the velocities are lower in the drain: the Gardon de Saint-Jean, experiment 1 ($P2$), the southwestern tributary connected to Corbès, experiment 1 ($P1$) and the upstream Gardon catchment in experiment 2 ($P2$). This is likely to be a consequence of an overestimated uniform background, with $\overline{v} = 5 \, \mathrm{m \, s^{-1}}$.

Calibration of the distributed hydrological models is a difficult task, with the data information content issue, the data interpretation criteria issue and the model adequacy. Some of these issues will be addressed in the near future. For example, the use of the Gaussian likelihood which leads to the quadratic cost function of the type in Eq. (15) does not seem to be the best choice. Taking into account that the discharge itself is a positive variable and the discharge observation error is, most likely, an increasing function of discharge, using the gamma likelihood looks more appropriate.

It is evident that, because of its conceptual nature and simplicity, the GRD model has some structural limitations. Looking for a simple structural upgrade, which may help to improve the adequacy without increasing noticeably the dimensions and computational costs, is an important future task. Another one is to provide better priors. For instance, one can use the nonuniform priors. In particular, for the capacities $c_p$ and $c_t$ one could consider locally uniform values relevant to the given hydrogeological or soil occupation class. For the velocity, a higher value in the drainage network with respect to the accumulated flow surface has to be enforced. The abovementioned subjects will be addressed in immediate future research.

In this paper, the model has been calibrated using data from different time periods, and the cross-validation experiments have helped to select the best optimal set of the distributed parameters. However, using a longer calibration period may not necessarily improve the model predictive performance due to the likely presence of very different hydrological regimes over the extended period, including some extreme cases. As a way forward, one could consider calibrating the model independently for different hydrological regimes. This approach is coherent to the idea of the "pooled analysis". Dependent on the calibration strategy, different parameter sets could be used for the prediction purpose. In the operational flood forecasting context, the model parameters are likely to be fixed, whereas the real-time update of the model initial state should be considered instead. In that case, the initial states of the distributed reservoirs will serve as a control vector, and the assimilation will be performed over a relatively short assimilation window (comparable to the characteristic time of the system). Finally, a running ensemble of models with different calibrated sets of parameters could be considered. In the framework of a flash flood warning system design, the latter approach could be combined with an automatic predictive control strategy such as the tree-based model (Ficchì et al., 2016).

In summary, the variational approach based on the adjoint model has proved its great computational efficiency and relevance for solving the calibration problem involving the distributed hydrological model GRD. Technically, this problem can be solved over long time periods and for large spatial areas. The difficulties discovered in this process are the fundamental issues of calibration, not related to the chosen method. The answer to the main research question formulated at the beginning of this paper is positive: it is possible and beneficial to calibrate and then use distributed parameters rather than uniform parameters. A variational algorithm involving adjoint sensitivities has proved to be an efficient tool for such calibration. The calibration quality is expected to be improved by using a more appropriate cost function and by enhancing the model structure. Overall, this means that the suggested research and hydrological forecasting tool development direction is quite promising.

*Data availability.* Meteorological and streamflow data used in this study can be obtained from Météo-France and the French ministry in charge of environment, respectively. The simulation data that support the findings of this study have been obtained using the SMASH (Spatially distributed Modellng and ASsimilation for Hydrology) platform coded in FORTRAN. They are available from the second author on reasonable request.

*Author contributions.* MJA performed the computational work. PJ, IG and PA supervised this work. All the co-authors collaborated, interpreted the results, wrote the paper and replied to the comments from the reviewers.

*Competing interests.* The authors declare that they have no conflict of interest.

*Special issue statement.* This article is part of the special issue "Hydrological cycle in the Mediterranean (ACP/AMT/GMD/HESS/NHESS/OS inter-journal SI)". It is not associated with a conference.

*Acknowledgements.* This research has been done in the framework of the 2010–2020 HyMex program. It contributes to the French national PICS (Prévision immédiate intégrée des impacts des crues soudaines) project. The authors wish to thank Etienne Leblois (IN-RAE Lyon) for having provided the flow direction map used in this study. The editor and the four anonymous referees are also thanked for their constructive comments and suggestions.

*Financial support.* This research has been supported by the "Agence nationale de la recherche" (ANR–17–CE03–0011), the Institut national de recherche en sciences et technologies pour l'environnement et l'agriculture (IRSTEA, now INRAE), and by the French ministry in charge of the environment.

*Review statement.* This paper was edited by Véronique Ducrocq and reviewed by four anonymous referees.

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

**Remarks from the typesetter**

TS1    According to our standards, changes like this must first be approved by the editor, as data have already been reviewed, discussed and approved. Please provide a detailed explanation for those changes that can be forwarded to the editor. Please note that this entire process will be available online after publication. Upon approval, we will make the appropriate changes. Thank you for your understanding.

TS2    Please see previous remark regarding editor's approval.

TS3    Please see previous remark regarding editor's approval.

TS4    Please see previous remark regarding editor's approval.

TS5    Please see previous remark regarding editor's approval.

TS6    Please see previous remark regarding editor's approval.

TS7    Please see previous remark regarding editor's approval.

TS8    Please see previous remark regarding editor's approval.

TS9    Please see previous remark regarding editor's approval.