# Peer review of "On the potential of variational calibration for a fully distributed hydrological model: application on a Mediterranean catchment"

_Hydrology and Earth System Sciences, 2019_

## Referee Comment (RC1) · Anonymous Referee #1 · 28 Nov 2019

I read with attention the paper proposed by Jay-Allemand et al. The issue of calibrating distributed hydrological models is of high interest for the readers of HESS. The authors apply a variational algorithm to estimate the parameters of a conceptual model distributed over a 1-km grid. Despite this very interesting and important topic, the authors failed in my opinion to produce an article publishable in HESS. Namely, I will point out in my review issues regarding the introduction and scientific objective of this work, the lack of in-depth analysis and discussion of the results, and the ill-posed mathematical problem that is studied.

[Figure]

Major remarks:

The paper lacks clear scientific questions. Three points are mentioned by the authors at the end of the introduction:

- upgrading the GRD model,

- calibrating GRD with a variational approach, and finally,

- upgrading the variational approach.

These three objectives are more similar to a model development process that is described in a project report than to science objectives tackled to fill a gap in knowledge in a research article. They are very specific to the chosen model and very general in terms of objectives (upgrading and calibrating are broad terms). They do not introduce valuable objectives for a scientific paper. Moreover, the proposed scientific objectives are not supported by a proper experiment protocol: the new GRD model is not compared to any benchmark (i.e. the former GRD model or another model), the calibration with the variational approach is not compared to a calibration with another approach, only with overly simple homogeneous parameter sets, and the improved variational algorithm is not compared with the classical variational approach. As a consequence, we don't know if the developments of this work are valuable for other research works, we just know their performance, with no landmark.

This lack of clear scientific questions comes with (in my opinion) a deficient introduction. Actually, an introduction aims at clarifying the work presented, through an introduction of the problem and proposed solution, a relevant literature review about what has been done and what is the gap of knowledge the authors want to address. These aspects are barely present in the introduction proposed by the authors. They start by presenting pros and cons of distributed models (raising the issue of equifinality, I acknowledge it, but not saying they are going to work on that somehow in this study), then they introduce DA methods used in hydrology, later on they introduce the concept

of variational approaches and some DA methods applied to hydrology. Then we have a small paragraph about the issue of calibration for distributed models, with only two references and none over the last 10 years. Finally the issue for flash floods forecasting and the goal of the paper are mentioned. There is no clear continuity in this sequence of topics not really connected together or to this study. For example, do we need 15 lines about DA methods used in hydrology for state updating while this study is not about that? I would recommend presenting a deep analysis of the literature regarding the calibration of distributed or semi-distributed hydrological models. If this is a challenge, then explain why, explain what has been tried before and to which extent what you propose in this article is new. As mentioned in the introduction, other distributed models exist. As a consequence, they are calibrated, some of them with sophisticated methods. Recent examples include Rakovec et al. (2016), Piniewski et al. (2017), among others. However, as they are not mentioned, we do not know how the work compares to these previous studies.

The abstract is written in a very unusual way. Usually, an abstract should contain some context, a description of the methodology, the most important results and finally one or two sentences about perspectives, in terms of further research or improvements. Here, the context is given, and then the rest of the abstract is about the methodology implemented. Only the last sentence provides some elements of results ("encouraging results") with no much detail, and perspectives are never provided. In my opinion, the abstract should be entirely rewritten.

Variational methods are powerful tools for data assimilation and parameter estimation, which are used for quite some time already in the fields of meteorology or oceanography. Their use in the field of hydrology is clearly less developed, especially compared to the EnKF or particle filter, but I highly doubt that "In hydrology, the variational estimation method as described above (i.e. including the adjoint model) has not been reported so far." (page 3, line 12 ; see also the conclusions "To the best of our knowledge, this is the first time when the variational estimation involving the adjoint sensitivities

has been applied in the field of hydrology."). For instance, in this journal, HESS, Castaings et al. (2009) seem to have developed a similar method. In addition, in their paper Castaings et al. cite some other works (see "Early applications of the adjoint state method to hydrological systems have been carried out in groundwater hydrology (Chavent, 1974; Carrera and Neuman, 1986; Sun and Yeh, 1990)" and the following sentences). Nguyen et al. (2016) might also be relevant. I encourage the authors to make a proper literature review on this specific aspect, which is necessary for the HESS audience to identify the added value of this specific study. In addition, the use of variational methods for data assimilation (i.e. state updating) is quite common in 'hydrology' in a broad sense.

The assessment of the performance is not developed. Only NSE values are calculated, while the authors specifically want to address flash floods. No criterion about peak-over-threshold, timing, intensity, is used. Since the study is already limited to a single watershed, limiting the analysis to a single criterion is out of the standards of nowadays hydrological studies.

The presentation of scores is poor. In figures 3, 4 and 5, the stations are ranked by their performance. It is a pity that we cannot identify anymore the stations. What a hydrologist would like would be to analyze whether there is a difference between experiments for a specific station, whether there are links between performances over upstream / downstream stations, etc. This kind of analyses is impossible to perform from the presented graphs. In addition, the fact that scores are mixed between the two periods (P1 and P2), if I understand well, is even more confusing.

As I mentioned at the beginning of my review, the analysis and discussion of the results is insufficient. The description of the results consists in a one-page long text and the discussion in less than 20 lines. Only NSE is presented to assess the performance, as well as the maps of parameters. This is a pity, as there would be a lot to say. First, it is clear from figures 6 and 7 that the parameter values are highly different when calibrated over P1 or P2. For many grid points, Cp and Ct can reach the lower bound for a period

and the upper bound for the other. The authors blame the change of precipitation between the periods or the chosen model. In my opinion, it simply indicates equifinality. Indeed, the performances do not decrease a lot between the periods but the parameter values are completely opposite. It comes from the fact that not enough information is provided to the algorithm to calibrate 540 x 3 parameter values. In other words, the optimization problem is ill posed. It indicates that calibrating these three parameters over each grid cell is not possible with only discharge time series and the variational algorithm. While the presentation of negative results is interesting and should be encouraged in my opinion, it has to come with a proper experiment protocol and in-depth analyses. If different precipitation patterns are the key factor for explaining these results, then the reader has no element to assess that: no mean or extreme precipitation values or even maps are provided.

The discussion of the results comes with several rude and coarse judgments, not supported by evidence. For instance, the authors blame "a structural deficiency of the chosen model" (page 16, line 1), saying it is not surprising since "the model is conceptual". Then, they state that "the hydrological modeling at the cell scale is very primitive". These two elements may explain why the parameter values are so different according to the authors. First, these assertions are very surprising, as the authors did develop the model they used, if I understand well. Second, being simple or conceptual is not necessarily a deficiency. On the opposite, it is often considered as being an advantage, as such models are easier to run or to understand. Third, if the authors identified a structural deficiency, then it has to be shown and analyzed, and solutions for improvement must be discussed. It is true that some processes are not modeled, but what shows that this is the reason of the poor results? In addition, the authors also suspect the "routing scheme" (line 3). I am surprised by that, as figure 7 and page 14, lines 29 to 30 indicate that the routing velocity is the best determined parameter.

Minor remarks:

The quality of the English used in this work is sometimes rather poor. The manuscript

contains a high number of formulation that are more typical from oral English than from written scientific English ("Let us note", "As we already said", etc.). I strongly recommend that a native English speaker reads and corrects the manuscript.

Page 1, lines 4 to 6: it is not clear whether AIGA is a forecasting system (as said on line 4) or not (as we learn on line 6 that it uses radar observations, not forecasts).

Page 1, line 8: "greater": do you mean "higher"?

Page 1, line 10: "have also" must be changed into "also have".

Page 1, line 10: "This must be larger enough": what do you mean? Do you mean "large enough"?

Page 3, lines 23 to 29: local methods indeed sometimes fail to identify global optima. However, between local methods and DA approaches, one can find the global optimization algorithms, which prove to be sometimes more efficient than local methods. DA is an option, not necessarily the only option!

Page 6; line4: I would say that the soil reaches its "minimal" absorption capacity when all rainfall contributes to runoff, rather than "maximal".

What is the meaning of the word "scalable" used in the introduction and in section 2.2?

Page 9, lines 14 to 19: this paragraph and equation is introduced, to finish by saying that it is not going to be used. This can be deleted not to confuse readers.

Page 10, line 2: this equation is not numbered.

Page 11, line 31: what is an "active" cell? Why mentioning the rectangular 1600 km$^2$ grid? GRD is not capable of running over catchments irregular shapes?

Page 12, line 1: this is a classical split sample test as defined by Klemes 30 years ago, it is worth mentioning it.

Page 12, line 8 to 9: it seems to me it is a proxy-basin test, as also defined by Klemes.

Please confirm.

Page 14, line 14: "One can see that the model spatial predictive performance is also better if the distributed calibration (red) is used, with one exception.": is that true? Since the stations are ranked, it might not be true, the reader cannot know.

Table 2 and 3 show a wrong unit for Ct. Indeed, if we check equation 8, then Ct must have the same unit as h and q, i.e. mm.

Page 15, the authors state that "For a chosen observation period and the associated test signal (rainfall) one can get a relatively stable set of calibrated parameters.". This statement is not supported by any kind of evidence in the manuscript and is not very clear. I guess that stability stands for temporal stability, but then how can it be assessed from a single period?

Page 17, line 23: is the code available under a GPL license? The proposed website requires a username but there is no possibility to register.

References:

Castaings, W., Dartus, D., Le Dimet, F.-X., and Saulnier, G.-M.: Sensitivity analysis and parameter estimation for distributed hydrological modeling: potential of variational methods, Hydrol. Earth Syst. Sci., 13, 503–517, https://doi.org/10.5194/hess-13-503-2009, 2009.

V. KLEMEŠ (1986) Operational testing of hydrological simulation models, Hydrological Sciences Journal, 31:1, 13-24, DOI: 10.1080/02626668609491024

Piniewski, M., Szcześniak, M., Kardel, I., Berezowski, T., Okruszko, T., Srinivasan, R., Schuler, D. V., & Kundzewicz, Z. W. (2017). Hydrological modelling of the Vistula and Odra river basins using SWAT. Hydrological Sciences Journal, 62(8), 1266–1289. https://doi.org/10.1080/02626667.2017.1321842

Rakovec, O., Kumar, R., Attinger, S., and Samaniego, L. (2016), Improving the real-

ism of hydrologic model functioning through multivariate parameter estimation, Water Resour. Res., 52, 7779– 7792, doi:10.1002/2016WR019430.

Van Tri Nguyen, Didier Georges, Gildas Besançon, Adjoint-based state and distributed parameter estimation in a switched hyperbolic overland flow model, IFAC-PapersOnLine, Volume 49, Issue 18, 2016, Pages 205-210, ISSN 2405-8963, https://doi.org/10.1016/j.ifacol.2016.10.164.

---

## Referee Comment (RC2) · Anonymous Referee #2 · 10 Dec 2019

The purpose of this article is to present a calibration methodology based on variational data assimilation using the adjoint state of a conceptual rainfall-runoff model. The proposed model is then tested in the Gardon catchment. The added value of the distributed version of the model is also studied.

The topic is of great interest in the field of hydrology and the proposition is valuable to make advances in operational calibration for rainfall-runoff modeling. However, the present article lacks: (i) a clear presentation of the novelty of the study in the general context of data assimilation using variational methods in hydrology or more broadly for

environmental model calibration and (ii) a more thorough and substantiated analysis of their results as detailed in the comments below.

1. The abstract should be more precise and synthetic, both on the context and on the results of the study.

2. P1 L5 Replace "accounting for spatial variability of the rainfall and the catchment properties, based on the radar rainfall observation inputs" by "accounting for spatial variability of the catchment properties and the rainfall, based on the radar rainfall observation inputs".

3. P1 L15 "scalable" has not been defined yet.

4. P3 L12 and P17 L9-10 The calibration of a distributed hydrological model using variational methods including the adjoint model has already been tested, at least several years ago in two PhD Thesis using the MARINE model (References below). I think a thorough and well-documented review of the state of the art research in this topic is needed to emphasize the novelty of the present study.

Bessière, H., Roux, H. and Dartus, D., 2007. Data assimilation and distributed flash flood modelling. Second Space for Hydrology Workshop - "Surface Water Storage and Runoff: Modeling, In-Situ data and Remote Sensing", Geneva, Switzerland. Available online http://earth.esa.int/hydrospace07/participants/08_01/08_01_Bessiere.pdf

Bessière, H., 2008. Assimilation de données variationnelle pour la modélisation hydrologique distribuée des crues à cinétique rapide. PhD thesis, Institut National Polytechnique de Toulouse, France. http://ethesis.inp-toulouse.fr/archive/00000710/

Castaings, W., 2009. Analyse de sensibilité et estimation de paramètres pour la modélisation hydrologique : potentiel et limitations des méthodes variationnelles. PhD thesis, Institut National Polytechnique de Grenoble, France. https://tel.archives-ouvertes.fr/tel-00264807

Castaings, W., Dartus, D., Le Dimet, F.-X., and Saulnier, G.-M.: Sensitivity analysis

and parameter estimation for distributed hydrological modeling: potential of variational methods, Hydrol. Earth Syst. Sci., 13, 503–517, https://doi.org/10.5194/hess-13-503-2009, 2009.

Larnier, K., Roux, H., Garambois, P.-A. and Dartus, D., 2012. Data assimilation method for real-time flash flood forecasting using a physically based distributed model, European Geosciences Union General Assembly 2012, Vienna, Austria. 14. Available online https://meetingorganizer.copernicus.org/EGU2012/EGU2012-12846.pdf

5. P4 L26 Replace "based on the temperature" with "based on the air temperature"

6. P6 L20 "This model is build on top of a digital elevation model, which defines the runoff directions between the routing nodes." Replace "build" with "built". How the runoff directions are defined? How many runoff directions are allowed for each cell: 4 or 8? Is there a fillsink algorithm?

7. P8 figure 1 The model doesn't include any representation of the surface flow? It is quite surprising for flash flood simulation. Even if the authors mentioned that for the Gardon catchment "the water circulation appends mainly underground" (P11 L20), this is likely not to be the case for all the catchments prone to flash flood. Moreover, even if the dominant process in the generation of runoff is not the Hortonian one, surface runoff can also be generated by soil saturation. Can the authors comment on this?

8. P9 L4 It is quite confusing to use the same letter P both for precipitation and for the parameter vector even if one is in capital letters and the other not.

9. P10 equation (20) the "T" is not in the right place, I assume it should be the limit of the integral.

10. P12 L6 "A model warm-up of one year long is performed before starting the simulations." Did you test the impact of the duration of the warm-up period on the simulation results? Is the initial state of both reservoirs completely "forgotten" after one year of simulation?

11. P13 L10-13 From a chronological point of view, it would be more relevant to calibrate on P2 and validate on P1 for forecasting purpose. Indeed, if there is a trend in the data, you will miss it when calibrating on P1 and validating on P2. Did you see any impact on your results?

12. P14 L15 "One can see that the model spatial predictive performance is also better if the distributed calibration (red) is used, with one exception" on fig. 4 right. There is also one exception on fig. 3 left (rank 8). Can the authors comment on these exceptions: any reasons? Maybe related to the catchment or the period? For the same reasons, it would be interesting to mention in the fig. 3 and 4 not the rank which is obvious but the catchment and the calibration period in order to see if there are any correlations between the performances and the calibration sets (also see previous comment on that aspect).

13. P14 L16 "This depends, however, on the spatial variability of the test signal (rainfall)." It would have been interesting to correlate the performances with the spatial variability of the rainfall, for instance using Zoccatelli et al. (2011) indices. This could also have helped justifying the analysis P14 L25 "This effect can be attributed to quite a different rainfall pattern over the reference periods."

Zoccatelli, D., Borga, M., Viglione, A., Chirico, G. B., and Blöschl, G.: Spatial moments of catchment rainfall: rainfall spatial organisation, basin morphology, and flood response, Hydrol. Earth Syst. Sci., 15, 3767–3783, https://doi.org/10.5194/hess-15-3767-2011, 2011.

14. P14 L28 "calibrating the model independently for different hydrological regimes" or maybe calibrating the model using a data set including the different hydrological regimes?

15. P16 Table 2 The parameter $c_t$ is presented as the capacity of the transfer reservoir (P4 L32) why isn't it in mm as $c_p$?

16. P16 Table 2 and P17 Table 3 are never mentioned in the text.

17. P16 L3-5 "This poor hydraulic behaviour can be partially compensated by the others model parameters during the calibration process, which may explain the extreme (equal to the upper bound) parameter values." I assume this refers to the value of 5 m/s for v in Table 2 and 3? What about 1000 mm for cp, is it the upper bound too? What does it mean for a physical point of view if you need a biggest production reservoir?

18. P16 L11 "hit" instead of "heat"?

19. It would have been interesting to study the correlation of the value of the calibrated parameters ct and cp with the actual soil properties: storage capacities but also soil texture and soil depths. Of course GRD is a conceptual model but the calibration of the local routing velocity v clearly shows a distinct behaviour of the drainage network. Is it the same for the 2 others parameters? Can the analysis of the correlation between ct, cp and the actual soil properties be instructive to improve the model structure or the calibration methodology? I think that could be one of the interesting contributions of the study.

---

## Referee Comment (RC3) · Anonymous Referee #3 · 12 Dec 2019

Review for manuscript HESS-2019-331

**"On the potential of variational calibration for a fully distributed model: application on a Mediterranean catchment"**

This study describes a new method based on variational data assimilation to calibrate a distributed hydrological model used operationally for flash flood forecasting in France. The method is applied on a French mountainous basin monitored by four stream gauge stations and rainfall radars. The authors first present the basics of the GRD distributed model and the new development that enables the differentiability of the model. Then the variational calibration method, the core of the study, is developed, as well as the methodology to evaluate it. The last two sections present the results and the discussion. Overall, the calibration strategy proposed by the authors is quite new and relevant for high dimensional model calibration, making this paper suitable for publication in HESS. Nevertheless, I have a few major comments I think the authors should address before publication. I would highly encourage the authors to further develop the evaluation of the methodology; the paper would then largely gain in visibility.

Major comments:

1. The methodology is quite well described but lacks for deeper evaluation. The results section is only one page and the discussion is reduced to less than 20 lines. I am sure that the study could benefit a lot from a more comprehensive analysis of this new methodology. For instance, some hydrographs could be included to show how the model behaves at the different stations depending on whether they are used or not in the calibration process. Also, do some stations always show good/bad performances? It lacks some time series to get an idea of interannual and spatial variability, flood severity (Qmax/Qmin).

2. Mathematical notations should be carefully revised as they are not always defined and sometimes not consistent (see following comments). This is very confusing and does not help the reader to fully understand how the method works.

Minor comments

P2L20. Do the authors know if there is any effort in the community to relate parameters of lumped or conceptual models to physical characteristics (e.g. average slope of the basin, concentration time...)? This could have important implication for the extension to ungauged basins.

P4L25. Is there any name or reference for the radar precipitation estimates provided by Météo-France?

P6. It appears to me that $q$ (from transfer function, Eq. (8)) represents the runoff, while $Q$ (from the routing model) represents the discharge in rivers. It is not clear in the text (L7 and L19).

P6L12. "Assuming Pn is the impulse function": isn't it Pr?

P6L22. Tau_i represents the runoff (or discharge? See previous comment) delay from node i-1 to node i. In a drainage network, a particular node i may have multiple direct upstream nodes i-1. Does Eq. (9) mean that all nodes i-1 flowing into node i share the same velocity and distance to node i? Rather, I would have defined v_i and d_i as the velocity in node i and the distance between node i and node i+1. Please correct me if I am wrong.

P6L22. N is not defined. Also, I would have removed "i=1,...,N" from the formula. Idem for Eq. (10)

P6L27. Variable q_i in Eq. (10) is not defined. I guess it is q at node i.

P7L15. K is not defined.

P7L20. What does "without the initial shock" mean?

P7L26. Identifiability is not only due to scarcity of observations in space. It may also comes from the model structure (concurrent parameters) and from processes (concurrent variables).

P8Fig1. I would suggest that variables that appear in the figure should be defined in the figure caption, even if they are defined in the text.

P9L1. Ns is not defined. Does it represent the number of nodes within the entire domain? Is it equal to N that appears previously (but not defined as well)?

P9L2. It is not clear from the formulation of Eq. (15) if Qk(t) only depends on P(xk,t),E(xk,t) or P and E over the entire domain. Actually, Qk(t), which represents the discharge at node xk and time t, should depend on P and E over the upstream sub-basin (which could be extended to the entire domain with a zero impact for nodes not inside the upstream sub-basin) and over a past period with a certain depth (which depends on the flow velocity of all the upstream cells, and which could be extended to the entire past period). Given that, I would have removed "k=1,Ns" from the equation and I would have changed t in the right hand side into t' with t' within the interval (0,t). Besides, in all the equations of section 2.1, i represents the node (discrete space) and k the time step (discrete time), which is not consistent with Eq. (15).

P9L3. h(x) should be h(x,0). Moreover, h(x,0) only represents the states of production and transfer reservoirs at node x, not over the entire domain. Same for p(x).

P10L1. The authors may explain that the scaling of p is done to get parameters within a [0,1] range.

P10L6. May p_tilde be equal to 0 or 1?

P10L7-11. Could the authors explain the implications of such assumptions?

P11L6. "Study area"

P11L27. "Investigating methodology"

P13L4. Letter "v" is missing in "validation". Please check this out throughout the section.

P14L9. Please show it in a new figure (e.g. time series of P averaged over the entire domain).

P14L11-12. Is this shown in Fig. 4?

P14L25-26. Could this be shown (see also previous comment)?

P16Table2. Is Table 2 cited in the text?

P16L1-2. Physically based model can also suffer from structural deficiencies.

P16L5. What justifies the given bound values? Would the results be better without these bounds? In my opinion, it is better not to constrain parameter values except in case of numerical constraints. This is especially true for a conceptual model for which parameter values have no interpretable physical meaning.

P17Table3. Is Table 3 cited in the text?

P17L1-2. Typically, this is a result that could be shown in a figure.

P17L5. In addition to the cumulative surface, the distributed parameters could be compared to land cover maps or soil texture maps. If any relationship appears, such data could be used to better constrain the calibration process and then limit equifinality issues.

P17L14-17. Please reformulate.

P23Figure6. Parameter values are very different whether P1 or P2 is considered as the calibration period. What are the implications on the stability of the calibration (similar performances with P1 or P2)?

---

## Author Comment (AC1) · 15 Jan 2020

Answer RC1,

We would like to thank the Reviewer for the careful reading. We gratefully consider all Reviewer's comments and suggestions.

1. The paper lacks clear scientific questions. Three points are mentioned by the authors at the end of the introduction: - upgrading the GRD model, - calibrating GRD with a variational approach, and finally, - upgrading the variational approach. These

three objectives are more similar to a model development process that is described in a project report than to science objectives tackled to fill a gap in knowledge in a research article. They are very specific to the chosen model and very general interms of objectives (upgrading and calibrating are broad terms). They do not introduce valuable objectives for a scientific paper. Moreover, the proposed scientific objectives are not supported by a proper experiment protocol: the new GRD model is not compared to any benchmark (i.e. the former GRD model or another model), the calibration with the variational approach is not compared to a calibration with another approach, only with overly simple homogeneous parameter sets, and the improved variational algorithm is not compared with the classical variational approach.

We agree with the reviewer that the scientific questions have not been explicitly formulated. In the original version of the AIGA model (which is the operational model currently utilized by SCHAPI) the mentioned above "overly simple homogeneous parameter set" is actually used. The main scientific question is: could we benefit from considering the spatially distributed set of coefficients instead of the uniform (homogeneous) set of coefficients and to what extent? In particular, does it help to improve the discharge prediction over the catchment area including 'ungauged' locations? To answer this question the predictive performance of the calibrated model involving spatially distributed coefficients against the one involving uniform coefficients has been performed. We specify more clearly this question in the introductory part. The GRD model comes from the GR family and shares the production and the transfer operators. The lumped GR models (GR4J) have been extensively tested in [Perrin C, 2000] and have shown good performance in modeling the hydrological processes. The distributed versions [De Lavenne A 2019 , Lobligeois F. 2014] take into account the spatially distributed rainfall observations at a refined scale. In this context the GRD model (upgraded AIGA model) designed especially for flood prediction has been developed. We acknowledge that no comparison between the upgraded AIGA model (employing the 'cell-to-cell' routing scheme) and the original version (employing the 'cell-to-outlet' routing scheme) is presented in this paper, but this is done to avoid overloading the paper with technical details. Notice that the cell-to-cell routing scheme is a requirement for our distributed model since it is designed to estimate the discharges at any location inside the watershed (involving ungauged sites), not only at the outlet.

2. Concerning suggested comparison "the calibration with the variational approach" "to a calibration with another approach".

The purpose of such comparison is not clear. Any MAP estimator (KF, EnKF, BLUE, etc.) would produce the same result (at least in theory). We assume that a Bayesian estimator (particle filter, MCMC, etc.), is not really feasible in high-dimensions (of course, it would be possible to go into different dimension reduction techniques, but this is not the direction we decided to follow). The useful comparison would probably be to some robust estimation methods, but this robust estimator is yet to be implemented.

3. Concerning "the improved variational algorithm is not compared with the classical variational approach".

Our statement about 'upgrading variational approach' mistakenly comes from our recent reports and is not relevant in the context of this paper which describes earlier results. Here we only use constrained minimization and scaling, but these features simply represent a 'good practice' approach in solving the calibration problems. So, this statement is relaxed in the introduction.

4. Concerning "These three objectives are more similar to a model development process that is described in a project report".

It depends. For example, in computational journals (JCP, CMAME, IJNMF, Comp. Geosciences, etc) this is a usual agenda in the majority of cases.

5. This lack of clear scientific questions comes with (in my opinion) a deficient introduction. The abstract is written in a very unusual way.

This remarks is consistent with reviewer 2. We agree that these parts should be improved. The introduction is rewritten to emphasis the objectives, the originality and the results of this study.

6. I highly doubt that In hydrology, the variational estimation method as described above (i.e. including the adjoint model) has not been reported so far. (page 3, line 12 ; see also the conclusions To the best of our knowledge, this is the first time when the variational estimation involving the adjoin sensitivities has been applied in the field of hydrology.). For instance, in this journal, HESS, Castaings et al. (2009) seem to have developed a similar method. In addition, in their paper Castaings et al. cite some other works (see Early applications of the adjoint state method to hydrological systems have been carried out in groundwater hydrology (Chavent, 1974; Carrera and Neuman, 1986; Sun and Yeh, 1990). Nguyen et al. (2016) might also be relevant. I encourage the authors to make a proper literature review on this specific aspect, which is necessary for the HESS audience to identify the added value of this specific study.

We agree with the reviewer and acknowledge that this is a significant flaw from our side. The reason for this 'amnesia' is, probably, that in these papers the hydrological problem is represented by the partial differential equations with a nonlinear source term: a standard formulation typical for classical problems in atmospheric science and oceanography. In our case, we introduced the delay-based 'cell-to-cell' conceptual routing scheme, which represents the difference rather than differential equations. Of course, saying "this is the first time when the variational estimation involving the adjoint sensitivities has been applied in the field of hydrology" is incorrect. There are some other papers besides the mentioned, but still, only a very few. Let us also note, that the adjoint of an operational geophysical model is always an exclusive piece of work, despite availability of the modern Automatic Differentiation tools. The bibliography is completed by the references suggested and some others. We highlight the differences with our work (model, objectives, methodology).

7. The assessment of the performance is not developed. Only NSE values are calculated, while the authors specifically want to address flash floods. No criterion about

peak-over-threshold, timing, intensity, is used. Since the study is already limited to a single watershed, limiting the analysis to a single criterion is out of the standards of nowadays hydrological studies.

We agree these remarks. First, the NSE criteria is not enough for assessing the performance since this criterion was used in calibration. We suggest to also use the KGE criteria. We agree that it makes sense to assess the performances over flood events with others criteria (timing). Our model runs continuously and such analysis would require splitting the discharge series and analyzing a selection of events. This work can be done and added to this paper.

8. The presentation of scores is poor. In figures 3, 4 and 5, the stations are ranked by their performance. It is a pity that we cannot identify anymore the stations. What a hydrologist would like would be to analyze whether there is a difference between experiments for a specific station, whether there are links between performances over upstream/downstream stations, etc. This kind of analyses is impossible to perform from the presented graphs. In addition, the fact that scores are mixed between the two periods (P1 and P2), if I understand well, is even more confusing.

The presentation chosen (results ranked by the value of NSE) was used to keep the analysis simple. While this representation could be suitable for analysis of a large set of catchments, we agree that for one catchment this may not be sufficient (only 4 stations and 2 periods). Thus, we provide a deeper analysis of the model performance by stations and by periods.

9. First, it is clear from figures 6 and 7 that the parameter values are highly different when calibrated over P1 or P2. For many grid points, Cp and Ct can reach the lower bound for a period C4 and the upper bound for the other. The authors blame the change of precipitation between the periods or the chosen model. In my opinion, it simply indicates equifinality. It comes from the fact that not enough information is provided to the algorithm to calibrate 540 x 3 parameter values.

We have exactly the same opinion. Equifinality manifests itself in the fact that for different 'test' signals we obtain quite different estimates of parameters. In the discussion part, we mention that equifinality issues may be reduced by introducing additional constraints to the minimization problem (spatial correlations, better background estimation, removing data outliers, etc.). This is done in more recent work. However one can notice the stability of the routing parameter "v" and some similarities of the others parameters between both periods (higher values at the North and South parts of the watershed).

10. In other words, the optimization problem is ill posed.

Equifinality means non-uniqueness, i.e. given the input, the same or nearly the same observed model output can be generated with different parameter sets. Regularization is used to chose one out of many possible solutions. On the contrary to what the Reviewer said, the calibration problem given by eq.(16) is well-posed. This does not help too much in practice, since the solution depends on the background $p\hat{a}\acute{L}\mathring{U}$ . The latter is obtained by considering the uniform coefficients (just 3 in total to calibrate: $C\grave{I}\check{D}\,p$ , $C\grave{I}\check{D}\,t$ , $v\grave{I}\check{D}$) and using the Monte-Carlo to build the joint posterior distribution, from which the mean values are taken as the background values. Even for these three coefficients we have different values of $C\grave{I}\check{D}\,p$ , $C\grave{I}\check{D}\,t$ , $v\grave{I}\check{D}$ for different periods $P\,1$ and $P\,2$ , see Tables 2,3. Thus, the problem is not due to over-parametrization.

11. It indicates that calibrating these three parameters over each grid cell is not possible with only discharge time series and the variational algorithm. While the presentation of negative results is interesting

What exactly is the criterion of successful calibration to assess whether the results are negative or positive? For the model which conceptually imitates the natural system, there exist no 'true set' of parameters. If the parameters are quantities of interest (QoI) by themselves, than the results are rather negative. If the QoI is the predicted state (discharge), it is rather positive, as have been shown using cross-validation. It is, of course, a comfortable feeling if for different inputs one could get consistent sets of

parameters, i.e. the estimates are stable, but is this so crucial for forecasting purpose ?

12. If different precipitation patterns are the key factor for explaining these results, then the reader has no element to assess that: no mean or extreme precipitation values or even maps are provided.

True, this will be provided. Particular events are studied (hydrograms and hydrographs are added). Moreover we would like to assess if the calibration (and the corresponding optimal solution) could be controlled by one particular event. For that, we compute a sliding NSE criteria over the calibration period and eventually detect if one particular event strongly influence the NSE.

12. The discussion of the results comes with several rude and coarse judgments, not supported by evidence. For instance, the authors blame a structural deficiency of the chosen model (page 16, line 1), saying it is not surprising since the model is conceptual. Then, they state that the hydrological modeling at the cell scale is very primitive. These two elements may explain why the parameter values are so different according to the authors. First, these assertions are very surprising, as the authors did develop the model they used, if I understand well. Second, being simple or conceptual is not necessarily a deficiency. On the opposite, it is often considered as being an advantage, as such models are easier to run or to understand. Third, if the authors identified a structural deficiency, then it has to be shown and analyzed, and solutions for improvement must be discussed.

"Structural deficiency" only means that the model (of a given structure) cannot be used to fit observations by manipulating its coefficients. This can be revealed by considering the 'forecast minus-observations' residuals. Despite a supposed over-parametrization (3 parameters for each cell), the residuals still contain some regular structures which cannot be explained solely by the data noise. While the presence of a structural deficiency is revealed, its correct interpretation and subsequent model improvement are

not easy. It could be that the reservoir models currently used are not perfectly suitable for the catchment under consideration, for example. It could be something else. Any discussion on this issue would be rather speculative at this stage. Some rewording of this part is done to avoid misunderstanding.

14. It is true that some processes are not modeled, but what shows that this is the reason of the poor results?

This is relevant to the discussion point raised above. What means 'the poor result' in our circumstances? Stability of estimates? Is it so vital to achieve this stability? In our view the results are promising: we get better discharge predictions at the gauged locations and, in average, in ungauged locations.

14. In addition, the authors also suspect the routing scheme (line 3). I am surprised by that, as figure 7 and page 14, lines 29 to 30 indicate that the routing velocity is the best determined parameter

This may be true, we agree with the reviewer.

All minor remarks are taken into account. Overall we agree that the presentation have to be improved and some additional materials have to be added in terms of results and their interpretation.

---

## Author Comment (AC3) · 15 Jan 2020

Answer RC3,

We would like to thank the Reviewer for careful reading. We gratefully consider all Reviewer's comments and suggestions (see below in italic).

Major comments :

1.The methodology is quite well described but lacks for deeper evaluation. The results

section is only one page and the discussion is reduced to less than 20 lines. I am sure that the study could benefit a lot from a more comprehensive analysis of this new methodology. For instance, some hydrographs could be included to show how the model behaves at the different stations depending on whether they are used or not in the calibration process. Also, do some stations always show good/bad performances? It lacks some time series to get an idea of interannual and spatial variability, flood severity (Qmax/Qmin).

These remarks are consistent with the remarks by Reviewers 1 and 2. Thus, we extend the analysis and the discussion. The extensions in progress are summarized below. 1- Previously, both calibration and validation have been performed using the same criterion (Nash-Sutcliffe (NSE) norm). Now, for validation we use both NSE and another criterion called KGE. This should provide a more reliable evaluation, because this criteria is different from the one used for calibration. 2- We analyze the results by periods and by stations as suggested by all Reviewers. 3- We focus on particular flood events and evaluate the model performance at stations using hydrograph and other event based criteria such as the temporal position of the modeled and observed flood peak, its intensity, etc. 4- More details are provided on the climatic properties of the Gardon watershed, such as the discharges quantiles at the gauge stations, the annual and seasonal average precipitation, the hourly rain intensities quantiles for different locations, etc. That helps to characterize the chosen events and to enhance the discussion.

2. Mathematical notations should be carefully revised as they are not always defined and sometimes not consistent (see following comments). This is very confusing and does not help the reader to fully understand how the method works.

We tried to combine classical notations from two different fields: hydrology and data assimilation. The notations are revised according to the Reviewer's comments.

Minor comments :

3. P2L20. Do the authors know if there is any effort in the community to relate parameters of lumped or conceptual models to physical characteristics (e.g. average slope of the basin, concentration time...)? This could have important implication for the extension to ungauged basins.

Many attempts have been made to relate the parameters of conceptual lumped models to the physical properties of watersheds. For example, one may cite [Lobliegeois Florent, IRSTEA, PhD Thesis, 2014]. In this work, the relationship between the parameters of the conceptual lumped model GR5H and catchment indicators are sought. It has been concluded that no obvious relationship exist for the capacities of production and transfer reservoirs. However, the routing scheme parameters are correlated with catchment properties such as slopes, surfaces and hydraulics distances. One can find some other publications on this subject, such as [Johansson, Barbro. "The relationship between catchment characteristics and the parameters of a conceptual runoff model: a study in the south of Sweden." IAHS Publications-Series of Proceedings and Reports-Intern Assoc Hydrological Sciences 221 (1994): 475-482.], [Post, David A., and Anthony J. Jakeman. "Predicting the daily streamflow of ungauged catchments in SE Australia by regionalising the parameters of a lumped conceptual rainfall-runoff model." Ecological Modelling 123.2-3 (1999): 91-104.], [Seibert, Jan. "Regionalisation of parameters for a conceptual rainfall-runoff model." Agricultural and forest meteorology 98 (1999): 279-293.], [Wagener, Thorsten, and Howard S. Wheater. "Parameter estimation and regionalization for continuous rainfall-runoff models including uncertainty." Journal of hydrology 320.1-2 (2006): 132-154.]. The common approach to solve this issue is called 'regionalization'. This approach is based either on relationship between parameters and catchment indicators, or on the spatial proximity of the catchments. One mat cite [Oudin, Ludovic, et al. "Spatial proximity, physical similarity, regression and ungaged catchments: A comparison of regionalization approaches based on 913 French catchments." Water Resources Research 44.3 (2008).], [Odry, Jean, and Patrick Arnaud. "Regionalisation of a distributed method for flood quantiles estimation: Revaluation of local calibration hypothesis to enhance the spatial structure

of the optimised parameter." EGU General Assembly Conference Abstracts. Vol. 18. 2016.], [Odry, Jean, and Patrick Arnaud. "Spatial disaggregation of a nationwide flood frequency analysis method." EGU General Assembly Conference Abstracts. Vol. 20. 2018.]. Unfortunately, the results presented in these papers are not generic, i.e. are relevant only to the models considered in these papers.

4. P4L25. Is there any name or reference for the radar precipitation estimates provided by Météo-France?

We use "ANTILOPE" radar rainfall product provided by Météo-France. This product corresponds to the re-analysis of the direct rainfall radar estimation with in-situ observations. This is to be mentioned in the revised version.

5. P6. It appears to me that q (from transfer function, Eq. (8)) represents the runoff, while Q (from the routing model) represents the discharge in rivers. It is not clear in the text (L7 and L19).

6. P6L12. "Assuming Pn is the impulse function": isn't it Pr?

We define notations more carefully.

7. P6L22. Tau_i represents the runoff (or discharge? See previous comment) delay from node i-1 to node i. In a drainage network, a particular node i may have multiple direct upstream nodes i-1. Does Eq. (9) mean that all nodes i-1 flowing into node i share the same velocity and distance to node i? Rather, I would have defined v_i and d_i as the velocity in node i and the distance between node I and node i+1. Please correct me if I am wrong.

You are correct.

8. P7L20. What does "without the initial shock" mean?

We should probably better explain this in the revised version. In Lag and Route method, the elementary discharge q(t) is a discontinuous function which is zero for t<t_0+delay.

It means the runoff from cell i arrives to cell i+1 as a 'shock'. If the Gaussian function is used, there is no discontinuity, i.e. runoff from cell i arrives to cell i+1 progressively. This scheme is more suitable for cell-to-cell implementation as it is more stable for direct modeling and it is absolutely necessary condition for the differentiability of the forward operator.

9. P7L26. Identifiability is not only due to scarcity of observations in space. It may also comes from the model structure (concurrent parameters) and from processes (concurrent variables).

This is true. We will mention this.

10.P16L5. What justifies the given bound values? Would the results be better without these bounds? In my opinion, it is better not to constrain parameter values except in case of numerical constraints. This is especially true for a conceptual model for which parameter values have no interpretable physical meaning.

The need for lower bounds is evident: neither reservoir capacities or velocity may be negative. Concerning the upper bounds, these are defined to preserve the model dynamics. For example, we choose 5 m/s as the velocity upper bound since above this value the system delay wont decrease significantly. For the production and transfer stores we set the upper bound to 1000 mm since higher values do not drastically change the model dynamics (reservoir remains close to steady state). In practice, if the upper bounds are very high (which is equivalent to the absence of such bounds), the calibration results are indeed better, however the validation results are noticeably worse.

12. P23Figure6. Parameter values are very different whether P1 or P2 is considered as the calibration period. What are the implications on the stability of the calibration (similar performances with P1 or P2)?

This question is not clear. If the Reviewer means 'What are the implications on the

stability of the validation' then the answer should be clear if the model performance is accessed as described in answers to major remark 1.

---

## Author Response (AR1)

Dear Editor,

Please find attached our revised version of our paper entitled "On the potential of variational calibration for a fully distributed hydrological model: application on a Mediterranean catchment"

We would like to thank you and the three anonymous reviewers for your constructive comments and suggestions.

Best regards,

On behalf of the authors

Hydrol. Earth Syst. Sci. Discuss.,
https://doi.org/10.5194/hess-2019-331

**Answers to reviewer #1**

We thank reviewer#1 for her/his constructive comments and suggestions.

| # | Reviewer comments | Authors answers | Change location in the new paper |
|---|---|---|---|
| 1 | Major remarks:The paper lacks clear scientific questions. Three points are mentioned by the authors at the end of the introduction:- upgrading the GRD model,- calibrating GRD with a variational approach, and finally,- upgrading the variational approach. These three objectives are more similar to a model development process that is de-scribed in a project report than to science objectives tackled to fill a gap in knowledge in a research article. They are very specific to the chosen model and very general in terms of objectives (upgrading and calibrating are broad terms). They do not introduce valuable objectives for a scientific paper. Moreover, the proposed scientific objectives are not supported by a proper experiment protocol: the new GRD model is not compared to any benchmark (i.e. the former GRD model or another model), the calibration with the variational approach is not compared to a calibration with another approach,only with overly simple homogeneous parameter sets, and the improved variational algorithm is not compared with the classical variational approach. As a consequence,we don't know if the developments of this work are valuable for other research works,we just know their performance, with no landmark. | New scientific objectives have been defined | P04L30-34 |
| 2 | This lack of clear scientific questions comes with (in my opinion) a deficient introduction. Actually, an introduction aims at clarifying the work presented, through an introduction of the problem and proposed solution, a relevant literature review about what has been done and what is the gap of knowledge the authors want to address. These aspects are barely present in the introduction proposed by the authors. They start by presenting pros and cons of distributed models (raising the issue of equifinality, I acknowledge it, but not saying they are going to work on that somehow in this study), then they introduce DA methods used in hydrology, later on they introduce the concept of variational approaches and some DA methods applied to hydrology. Then we have a small paragraph about the issue of calibration for distributed models, with only two references and none over the last 10 years. Finally the issue for flash floods forecasting and the goal of the paper are mentioned. There is no clear continuity in this sequence of topics not really | Introduction has been entirely re-written | P01P04 |

| | | | |
|---|---|---|---|
| | connected together or to this study. For example, do we need 15 lines about DA methods used in hydrology for state updating while this study is not about that? | | |
| 3 | I would recommend presenting a deep analysis of the literature regarding the calibration of distributed or semi-distributed hydrological models. If this is a challenge, then explain why, explain what has been tried before and to which extent what you propose in this article is new. As mentioned in the introduction, other distributed models exist. As a consequence, they are calibrated, some of them with sophisticated methods. If this is a challenge, then explain why, explain what has been tried before and to which extent what you propose in this article is new. As mentioned in the introduction, other distributed models exist. As a consequence, they are calibrated, some of them with sophisticated methods. | The novelty is the use of a VA for calibrating over a continuous long period with a split sample test (2x5years) | P04L25-30 |
| 4 | Recent examples include Rakovec et al. (2016), Piniewski et al. (2017),among others. However, as they are not mentioned, we do not know how the work compares to these previous studies | We ve completed the bibliography related to distributed calibration | P02L23-31 |
| 5 | The abstract is written in a very unusual way. Usually, an abstract should contain some context, a description of the methodology, the most important results and finally one or two sentences about perspectives, in terms of further research or improvements. Here, the context is given, and then the rest of the abstract is about the methodology implemented. Only the last sentence provides some elements of results ("encouraging results") with no much detail, and perspectives are never provided. In my opinion, theabstract should be entirely rewritten. | The abstract has been entirely rewritten | L01_P01-17 |
| 6 | Variational methods are powerful tools for data assimilation and parameter estimation,which are used for quite some time already in the fields of meteorology or oceanography. Their use in the field of hydrology is clearly less developed, especially compared tothe EnKF or particle filter, but I highly doubt that "In hydrology, the variational estimation method as described above (i.e. including the adjoint model) has not been reported so far." (page 3, line 12 ; see also the conclusions "To the best of our knowledge,this is the first time when the variational estimation involving the adjoint sensitivities. For instance, in this journal, HESS, Castaings et al. (2009) seem to have developed a similar method. In addition, in their paper Castaings et al. cite some other works (see "Early applications of the adjoint state method to hydrological systems have been carried out in groundwater hydrology (Chavent, 1974; Carrera and Neuman, 1986; Sun and Yeh, 1990)" and the following sentences). Nguyen et al. (2016) might also be relevant. I encourage the authors to make a proper literature review on this specific aspect, which is necessary for the HESS audience to | We ve completed the bibliography related to use of variational approach in hydrology | P04L09-21 |

| | | | |
|---|---|---|---|
| | identify the added value of this specific study. In addition, the use of variational methods for data assimilation (i.e. state updating) is quite common in 'hydrology' in a broad sense. | | |
| 7 | The assessment of the performance is not developed. Only NSE values are calculated,while the authors specifically want to address flash floods. No criterion about peak-over-threshold, timing, intensity, is used. Since the study is already limited to a single watershed, limiting the analysis to a single criterion is out of the standards of nowadays hydrological studies | New criterias were added | P15L14-P16L04 |
| 8 | The presentation of scores is poor. In figures 3, 4 and 5, the stations are ranked by their performance. It is a pity that we cannot identify any more the stations. What a hydrologist would like would be to analyze whether there is a difference between experiments for a specific station, whether there are links between performances over upstream / downstream stations, etc. This kind of analyses is impossible to perform from the presented graphs. In addition, the fact that scores are mixed between the two periods (P1 and P2), if I understand well, is even more confusing. | New paragraph and new fig4 were added to do this analyse | P17L01-P20L11 |
| 9 | As I mentioned at the beginning of my review, the analysis and discussion of the results is insufficient. The description of the results consists in a one-page long text and the discussion in less than 20 lines. Only NSE is presented to assess the performance, as well as the maps of parameters. This is a pity, as there would be a lot to say. First, it is clear from figures 6 and 7 that the parameter values are highly different when calibrated over P1 or P2. For many grid points, Cp and Ct can reach the lower bound for a period and the upper bound for the other. The authors blame the change of precipitation between the periods or the chosen model. In my opinion, it simply indicates equifinality. Indeed, the performances do not decrease a lot between the periods but the parameter values are completely opposite. It comes from the fact that not enough information is provided to the algorithm to calibrate 540 x 3 parameter values. In other words, the optimization problem is ill posed. It indicates that calibrating these three parameters over each grid cell is not possible with only discharge time series and the variational algorithm. While the presentation of negative results is interesting and should be encouraged in my opinion, it has to come with a proper experiment protocol and in depth analyses. | 2 new paragraphs were added to discuss that point | P17L01-P22L09 |
| 10 | If different precipitation patterns are the key factor for explaining these results, then the reader has no element to assess that: no mean or extreme precipitation values or even maps are provided. | This has been removed | |
| 11 | The discussion of the results comes with several rude and coarse judgments, not supported by evidence. For instance, the authors blame "a structural deficiency of the chosen model" (page 16, line 1), saying it is not surprising since "the model is conceptual". Then, they state that "the hydrological modeling at the cell scale is | You are right. This has been mentioned as future work | P24L22-24 |

| | | | |
|---|---|---|---|
| | very primitive". These two elements may explain why the parameter values are so different according to the authors. First, these assertions are very surprising, as the authors did develop the model they used, if I understand well. Second, being simple or conceptual is not necessarily a deficiency. On the opposite, it is often considered as being an advantage, as such models are easier to run or to understand. Third, if the authors identified a structural deficiency, then it has to be shown and analyzed, and solutions for improvement must be discussed. It is true that some processes are not modeled, but what shows that this is the reason of the poor results? In addition, the authors also suspect the "routing scheme" (line 3). I am surprised by that, as figure 7 and page 14,lines 29 to 30 indicate that the routing velocity is the best determined parameter. | | |
| 12 | The quality of the English used in this work is sometimes rather poor. The manuscript contains a high number of formulation that are more typical from oral English than from written scientific English ("Let us note", "As we already said", etc.). I strongly recommend that a native English speaker reads and corrects the manuscript | English has been corrected | |
| 13 | Page 1, lines 4 to 6: it is not clear whether AIGA is a forecasting system (as said online 4) or not (as we learn on line 6 that it uses radar observations, not forecasts). | This has been removed | |
| 14 | Page 1, line 8: "greater": do you mean "higher"? | This has been removed | |
| 15 | Page 1, line 10: "have also" must be changed into "also have". | This has been removed | |
| 16 | Page 1, line 10: "This must be larger enough": what do you mean? Do you mean "large enough"? | This has been removed | |
| 17 | Page 3, lines 23 to 29: local methods indeed sometimes fail to identify global optima. However, between local methods and DA approaches, one can find the global optimization algorithms, which prove to be sometimes more efficient than local methods. DA is an option, not necessarily the only option! | Global optimisation has been mentioned | P04L06 |
| 18 | Page 6; line4: I would say that the soil reaches its "minimal" absorption capacity when all rainfall contributes to runoff, rather than "maximal". | This has been removed | |
| 19 | What is the meaning of the word "scalable" used in the introduction and in section 2.2? | scalable has been defined | P03L19 |
| 20 | Page 9, lines 14 to 19: this paragraph and equation is introduced, to finish by saying that it is not going to be used. This can be deleted not to confuse readers. | We did not remove this paragraph, because usually DA is used in this way in flood | |

| | | forecasting | |
|---|---|---|---|
| 21 | Page 10, line 2: this equation is not numbered. | we prefer to leave as it is | |
| 22 | Page 11, line 31: what is an "active" cell? Why mentioning the rectangular 1600 km2grid? GRD is not capable of running over catchments irregular shapes? | "active cell" has been defined | P13L14 |
| 23 | Page 12, line 1: this is a classical split sample test as defined by Klemes 30 years ago,it is worth mentioning it. | We added the paper from Klemes. Thank you. | P14L01 |
| 24 | Page 14, line 14: "One can see that the model spatial predictive performance is also better if the distributed calibration (red) is used, with one exception.": is that true? Since the stations are ranked, it might not be true, the reader cannot know | This has been removed | |
| 25 | Table 2 and 3 show a wrong unit for Ct. Indeed, if we check equation 8, then Ct must have the same unit as h and q, i.e. mm. | It is now tab3 | P20L00 |
| 26 | Page 15, the authors state that "For a chosen observation period and the associated test signal (rainfall) one can get a relatively stable set of calibrated parameters.". This statement is not supported by any kind of evidence in the manuscript and is not very clear. I guess that stability stands for temporal stability, but then how can it be assessed from a single period? | This has been removed | |
| 27 | Page 17, line 23: is the code available under a GPL license? The proposed websiterequires a username but there is no possibility to register. | This has been removed | |

Hydrol. Earth Syst. Sci. Discuss.,
https://doi.org/10.5194/hess-2019-331

**Answers to reviewer #2**

We thank reviewer#2 for her/his constructive comments and suggestions.

| # | Reviewer #2 comments | Authors answers | Change location in the new paper |
|---|---|---|---|
| 1 | The present article lacks: (i) a clear presentation of the novelty of the study in the general context of data assimilation using variational methods in hydrology or more broadly fo environmental model calibration | The novelty is the use of a VA for calibrating over a continuous long period with a split sample test (2x5years) | P04L25-30 |
| | (ii) a more thorough and substantiated analysis of their results as detailed in the comments below | New analysis are provided (see below) | |
| 2 | 1. The abstract should be more precise and synthetic, both on the context and on the results of the study | The abstract has been entirely rewritten | L01_P01-17 |
| 3 | 2. P1 L5 Replace "accounting for spatial variability of the rainfall and the catchment properties, based on the radar rainfall observation inputs" by "accounting for spatial variability of the catchment properties and the rainfall, based on the radar rainfall observation inputs" | This has been removed | |
| 4 | 3. P1 L15 "scalable" has not been defined yet. | Removed from the abstract but defined later see P03L19 | P03L19 |
| 5 | 4. P3 L12 and P17 L9-10 The calibration of a distributed hydrological model using variational methods including the adjoint model has already been tested, at least several years ago in two PhD Thesis using the MARINE model (References below). I think a thorough and well-documented review of the state of the art research in this topic is needed to emphasize the novelty of the present study | We ve completed the bibliography related to distributed calibration | P02L23-31 |
| 6 | idem | We ve completed the bibliography related to use of variational | P04L09-21 |

| | | approach in hydrology | |
|---|---|---|---|
| 7 | 5. P4 L26 Replace "based on the temperature" with "based on the air temperature" | done | P13L02 |
| 8 | 6. P6 L20 Replace "build" with "built". | done | P07L09 |
| 9 | 6. P6 L20 "This model is build on top of a digital elevation model, which defines therunoff directions between the routing nodes.". How the runoff directions are defined? How many runoff directions are allowed for each cell: 4or 8? Is there a fill sink algorithm? | We have 8 directions | P13L06 |
| 10 | 7. P8 figure 1 The model doesn't include any representation of the surface flow? It is quite surprising for flash flood simulation. Even if the authors mentioned that for the Gardon catchment "the water circulation appends mainly underground" (P11 L20), this is likely not to be the case for all the catchments prone to flash flood. Moreover, even if the dominant process in the generation of runoff is not the Hortonian one, surface runoff can also be generated by soil saturation. Can the authors comment on this? | We added some sentences explaining that the GR models don't represent explicitly the hydrological processes | P05L26-29 |
| 11 | 8. P9 L4 It is quite confusing to use the same letter P both for precipitation and for theparameter vector even if one is in capital letters and the other not | p has been changed in teta anywhere | P09L12 |
| 12 | 9. P10 equation (20) the "T" is not in the right place, I assume it should be the limit of the integral. | done | P11L02 |
| 13 | 10. P12 L6 "A model warm-up of one year long is performed before starting the simu-lations." Did you test the impact of the duration of the warm-up period on the simulation results? Is the initial state of both reservoirs completely "forgotten" after one year of simulation? | No, but this has been verified by Perrin et al (2003) | P14L04 |
| 14 | 11. P13 L10-13 From a chronological point of view, it would be more relevant to calibrate on P2 and validate on P1 for forecasting purpose. Indeed, if there is a trend in the data, you will miss it when calibrating on P1 and validating on P2. Did you see any impact on your results? | New paragraph added and new fig4 | P17L01-P20L11 |
| 15 | 12. P14 L15 "One can see that the model spatial predictive performance is also better if the distributed calibration (red) is used, with one exception" on fig. 4 right. There is also one exception on fig. 3 left (rank 8). Can the authors comment on these exceptions:any reasons? Maybe related to the catchment or the period? For the same reasons,it would be interesting to mention in the fig. 3 and 4 not the rank which is obvious but the catchment and the calibration period in order to see if there are any correlations between the performances and the calibration sets (also see previous comment on that aspect). | New paragraph and new fig4 were added to do this analyse | P17L01-P20L11 |
| 16 | 13. P14 L16 "This depends, however, on the spatial variability of the test signal (rain-fall)." It would have been interesting to correlate the performances with the spatial variability of the | This has been removed | |

| | | | |
|---|---|---|---|
| | rainfall, for instance using Zoccatelli et al. (2011) indices. This could also have helped justifying the analysis P14 L25 "This effect can be attributed to quite a different rainfall pattern over the reference periods." | | |
| 17 | 14. P14 L28 "calibrating the model independently for different hydrological regimes"or maybe calibrating the model using a data set including the different hydrological regimes? | This has been removed | |
| 18 | 15. P16 Table 2 The parameter ct is presented as the capacity of the transfer reservoir(P4 L32) why isn't it in mm as cp? | It is now tab3 | P20L00 |
| 19 | P16 Table 2 never mentioned in the text. | It is now tab3 | P20L01 |
| 20 | P17 Table 3 are never mentioned in the text. | done | P20L01 |
| 21 | 17. P16 L3-5 "This poor hydraulic behaviour can be partially compensated by the others model parameters during the calibration process, which may explain the extreme (equal to the upper bound) parameter values." I assume this refers to the value of 5 m/s for v in Table 2 and 3? What about 1000 mm for cp, is it the upper bound too? What does it mean for a physical point of view if you need a biggest production reservoir? | 2 new paragraphs has been added to discuss that point | P17L01-P22L09 |
| 22 | 18. P16 L11 "hit" instead of "heat"? | This has been removed | |
| 23 | 19. It would have been interesting to study the correlation of the value of the calibrated parameters ct and cp with the actual soil properties: storage capacities but also soil texture and soil depths. Of course GRD is a conceptual model but the calibration of the local routing velocity v clearly shows a distinct behaviour of the drainage network. Is it the same for the 2 others parameters? Can the analysis of the correlation between ct, cp and the actual soil properties be instructive to improve the model structure or the calibration methodology? I think that could be one of the interesting contributions of the study. | You are right. This has been mentioned as future work | P24L24-28 |

Hydrol. Earth Syst. Sci. Discuss.,
https://doi.org/10.5194/hess-2019-331

**Answers to reviewer #3**

We thank reviewer#3 for her/his constructive comments and suggestions.

| # | Reviewer comments | Authors answers | Change location in the new paper |
|---|---|---|---|
| 1 | The methodology is quite well described but lacks for deeper evaluation. For instance, some hydrographs could be included to show how the model behaves at the different stations depending on whether they are used or not in the calibration process | A new section and Fig 8 analyse now some hydrographs | P18L17- and P34 |
| 2 | do some stations always show good/bad performances? | New paragraph and new fig4 were added to do this analyse | P17L01-P20L11 |
| 3 | It lacks some time series to get an idea of interannual and spatial variability, flood severity (Qmax/Qmin) | To have an idea of the intensity of the observed peak flood 2-y and 10-y flood quantiles were added in tab1 | P14L00 |
| 4 | P2L20. Do the authors know if there is any effort in the community to relate parameters of lumped or conceptual models to physical characteristics (e.g. average slope of the basin, concentration time...)? This could have important implication for the extension to ungauged basins. | We ve completed the bibliography related to distributed calibration | P02L23-31 |
| 5 | P4L25. Is there any name or reference for the radar precipitation estimates provided by Météo- France? | The name is ANTILOPE. This has been added | P13L01 |
| 6 | P6. It appears to me that q (from transfer function, Eq. (8)) represents the runoff, while Q (from the routing model) represents the discharge in rivers. It is not clear in the text (L7 and L19). | We tried to make clear the distinction between q and Q. | P07L17-18 |
| 7 | P6L12. "Assuming Pn is the impulse function": isn't it Pr? | Done / thank you | P07L01 |
| 8 | P6L22. Tau_i represents the runoff (or discharge? See previous comment) delay from node i-1 to node i. In a drainage network, a particular node i may have multiple direct upstream nodes i-1. Does Eq. (9) mean that all nodes i-1 flowing into node i share the same velocity and distance to node i? Rather, I would have defined v_i and d_i as the velocity in node i and the distance | You are right. Thank you. | P07L13-15 |

| | | | |
|---|---|---|---|
| | between node i and node i+1. Please correct me if I am wrong. | | |
| 9 | P6L22. N is not defined. | This has been removed | |
| 10 | P6L22. Also, I would have removed "i=1,...,N" from the formula. Idem for Eq. (10) | This has been removed | |
| 11 | P6L27. Variable q_i in Eq. (10) is not defined. I guess it is q at node i. | Yes. It is now defined | P07L17-18 |
| 12 | define K | This has been removed | |
| 13 | P7L20. What does "without the initial shock" mean? | This has been added | P08L14-18 |
| 14 | P7L26. Identifiability is not only due to scarcity of observations in space. It may also comes from the model structure (concurrent parameters) and from processes (concurrent variables). | You are right. This has been added | P08L23 |
| 15 | P8Fig1. I would suggest that variables that appear in the figure should be defined in the figure caption, even if they are defined in the text. | This has been added | P09L00 |
| 16 | P9L1. Ns is not defined. Does it represent the number of nodes within the entire domain? Is it equal to N that appears previously (but not defined as well)? | Ns was renamed in Ng and defined | P09L13 |
| 17 | P9L2. It is not clear from the formulation of Eq. (15) if Qk(t) only depends on P(xk,t),E(xk,t) or P and E over the entire domain. Actually, Qk(t), which represents the discharge at node xk and time t, should depend on P and E over the upstream sub-basin (which could be extended to the entire domain with a zero impact for nodes not inside the upstream sub-basin) and over a past period with a certain depth (which depends on the flow velocity of all the upstream cells, and which could be extended to the entire past period). Given that, I would have removed "k=1,Ns" from the equation and I would have changed t in the right hand side into t' with t' within the interval (0,t). | This has been corrected. Thank you | P09L10-11 |
| 18 | Besides, in all the equations of section 2.1, i represents the node (discrete space) and k the time step (discrete time), which is not consistent with Eq. (15). | k has been changed in l in (13) | P08L04-10 |
| 19 | P9L3. h(x) should be h(x,0). Moreover, h(x,0) only represents the states of production and transfer reservoirs at node x, not over the entire domain. Same for p(x). | done | P09L10-11 |
| 20 | P10L1. The authors may explain that the scaling of p is done to get parameters within a [0,1] range. | done | P10L24 |
| 21 | P10L6. May p_tilde be equal to 0 or 1? | yes. We made the change accordingly | P10L24 |
| 22 | P10L7-11. Could the authors explain the implications of such assumptions? | We added some explanations | P11L09-11 |

| | | | |
|---|---|---|---|
| 23 | P11L6. "Study area" | changed | P11L12 |
| 24 | P11L27. "Investigating methodology" | changed | P13L11 |
| 25 | P13L4. Letter "v" is missing in "validation". Please check this out throughout the section. | corrected | |
| 26 | P14L9. Please show it in a new figure (e.g. time series of P averaged over the entire domain). | This has been removed | |
| 27 | P14L11-12. Is this shown in Fig. 4? | comment added | P16L23-24 |
| 28 | P14L25-26. Could this be shown (see also previous comment)? | This has been removed | |
| 29 | P16Table2. Is Table 2 cited in the text? | It is now tab3 | P20L01 |
| 30 | P16L1-2. Physically based model can also suffer from structural deficiencies. | You are right. This has been mentioned as future work | P24L22-24 |
| 31 | P16L5. What justifies the given bound values? Would the results be better without these bounds? In my opinion, it is better not to constrain parameter values except in case of numerical constraints. This is especially true for a conceptual model for which parameter values have no interpretable physical meaning. | This information was added | P12L03-09 |
| 32 | P17Table3. Is Table 3 cited in the text? | done | P20L01 |
| 33 | P17L1-2. Typically, this is a result that could be shown in a figure. | new Fig 8 | P34 |
| 34 | P17L5. In addition to the cumulative surface, the distributed parameters could be compared to land cover maps or soil texture maps. If any relationship appears, such data could be used to better constrain the calibration process and then limit equifinality issues. | You are right. This has been mentioned as future work | P24L24-28 |
| 35 | P17L14-17. Please reformulate. | removed | |
| 36 | P23Figure6. Parameter values are very different whether P1 or P2 is considered as the calibration period. What are the implications on the stability of the calibration (similar performances with P1 or P2)? | 2 new paragraphs has been added to discuss that point | P17L01-P22L09 |

[revised manuscript text omitted]

---

## Referee Report (RR1)

[referee-annotated manuscript omitted]

---

## Author Response (AR2)

Dear Editor,

Please find attached our revised version of our paper entitled "On the potential of variational calibration for a fully distributed hydrological model: application on a Mediterranean catchment"

We would like to thank you and the three anonymous reviewers for your constructive comments and suggestions.

Best regards,

On behalf of the authors

Hydrol. Earth Syst. Sci. Discuss.,
https://doi.org/10.5194/hess-2019-331

**Answer to reviewer #1**

We thank reviewer #1 for her/his constructive comments and suggestions.

| # | Reviewer comments | Authors answers | Change location in the new paper |
|---|---|---|---|
| 1 | *The manuscript still lacks a short discussion of the applicability of the method in an operational context. For example, if final performances highly depend on the observed data used for the calibration (very different sets of optimized parameters), what are the consequences for operational flood forecast? What would be the best strategy (running an ensemble of models with different sets of parameters calibrated over different periods, calibration over all the available observations, etc.)? Could another experiment (calibration over the whole available period) help answering this question?* | We agree with the reviewer. Those questions have been answered into a new paragraph , in the "Discussion and Conclusions" part. | P25L9-21 |
| 2 | *P16L16: "This result is expected", it suggests that hydrological processes are not uniform over the catchment and that the spatial variability may be captured using distributed parameters.* | We've changed the sentence according to the reviewer #1 suggestion. | P16L16-18 |
| 3 | *P17L3: "Two of them (in "×"))": do these markers correspond to the Anduze gauge station which is used for calibration?* | Yes these markers show performance at the Anduze gauge station. This gauge is used for calibration but performance are presented for the validation period (different that the one used for calibration). Thus, for the validation period, we distinguish performance at the downstream gauge (temporal validation, two points corresponding to period P1 and P2 at the calibration station) and at the upstream gauges (spatio-temporal validation, 4x2 points corresponding to period P1 and P2 at the validation stations). We've | P17L2-P18L1 |

| | | | |
|---|---|---|---|
| | | slightly changed the original formulation in the text. | |
| 4 | *Fig. 8: subplot title (Event 2/P1)* | This subplot title has been corrected ("V7124015" instead of *"Event 2/P1"*). | P32 |
| 5 | *The figures and table numbering should be revised: Fig 8 should be Fig 5, Tab 5 and Tab 6 should be Tab 3 and Tab 4* | Position and numbering of figures and tabular have been modified according the reviewer #1 comment. | |

Hydrol. Earth Syst. Sci. Discuss.,
https://doi.org/10.5194/hess-2019-331

**Answer to reviewer #2**

We thank reviewer #2 for her/his constructive comments and suggestions.

| # | Reviewer comments | Authors answers | Change location in the new paper |
|---|---|---|---|
| 1 | *What are the main take-home messages of these studies? What hasn't been tested that you are going to test with the present study? (P4L13-25)* | In this paper, the parameters of a distributed model are calibrated, using a variationnal method which requires the adjoin model sensitivities. This calibration requires a very long assimilation period (several years). We tested the ability of this algorithm to assess the spatial variability of the hydrological watershed behavior using local discharge and spatial rainfall observations. To our knowledge, this has never be done in hydrology. We've tried to make that clearer in the introduction and also in the part describing the variationnal calibration algorithm. | P4L27-29 and P10L22 |
| 2 | *..the amount of net... (P6L24)* | This has been corrected. | P6L25 |
| 3 | omega before?*(P8L12)* | The equation is correct. \Beta are the coefficients computed according the \omega function defined at equation 12. | P8L15 |
| 4 | *alpha is already used in equation 6.* *beta is already used in equation 13 (equation 20)* | We replace \alpha with the letter "a" in equation 6. We replace \beta by \gamma in equation 20. | P7L8-10 P11L21 |
| 5 | *Impact on simulations? (P8L15)* | This coefficient impacts the spread (i.e diffusion). For sigma=0.5 and at the hourly time step, discharge spreads at least between two/three times-steps so that a small variation of the routing velocity introduces a rather small variation of the delay. Lower value of this coefficient introduces some numerical instabilities into cascade flow | P8L18-20 |

| | | | propagation scheme. It is a limitation of this model. A better description of the rule of this coefficient has been added. | |
|---|---|---|---|---|
| 6 | *Not needed and confusing as only equation 15 is used. (P10L16-21)* | | We agree with the reviewer #2 that this equation can be confusing. We removed these lines. | P10L21-22 |
| 7 | *O has never been mentioned before? (P11L10)* | | In the standard Tickhonov cost function, O stands for the observation errors matrix. This matrix is not used in this paper (O=I) and was mistakenly described here. | P11L11 |
| 8 | *simplifications mentioned above instead of mentioned above simplification (P10L14)* | | This has been corrected according to the reviewer suggestion. | P11L15 |
| 9 | *Impact on the results? (P12L15)* | | In these experiments, the upper bound are kept very large. Empirical calibration experiments have shown slightly better model performances with larger bounds for this watershed. Larger bounds allow higher deviation of the parameters from the background. Smaller bounds will constraint the parameters variability. We precise this idea in the text. | P12L15-16 |
| 10 | *Please choose between Gardon d'Anduze and Gardon of Anduze (P12L18)* | | We choose the formulation Gardon d'Anduze. | |
| 11 | *Calibration instead of validation? (P15L16)* | | No, this is correct. The experiment 2 evaluates the model performance at two levels : 1- performance at the validation catchments using data of the calibration periods. 2- performances at the validation catchments using data of the validation period. | |
| 12 | *Not always: it's not the case for Rank 1. What is the corresponding station/simulation? This needs to be discussed. (P18L1)* | | Yes it is not always the case, but this have been extensively discussed in the section Period-Based analysis. We modified slightly the sentence. | P17L5 |
| 13 | *Any idea why? Different event characteristics? (P18L34)* | | There is no significant change in event characteristics for two periods. We also checked if the calibration on period P1 was dominated by one particular (extreme) event, but without success. So, it remains an open question. | |

| 14 | *But the values are very different in fig. 5 (P20L13-14)* | It's true that the values are different. The velocity change occurs mainly at the cells located in the drainage, because the sensitivity of the cost function w.r.t. velocities in these cells is the largest. The trend is that these velocities increase. Note that velocities above 5 m/s does not affect significantly the delay between cells. We changed the original results description. | P20L15-17 |
|----|----|----|----|
| 15 | *Or physical characteristics? Dealing with extreme flood events, 5 years periods may not be statistically representative (P20L16-18) and (P23L10-11)* | Or physical characteristics, indeed. The corresponding sentence is modified. | P21L5 |
| 16 | *I'm not convinced: data selection is a part of the data processing! (P24L9)* | By saying this we meant that we have to use available data and we cannot add additional sensors, for example. However, data selection from all available data could be indeed a useful step. | P24L29 |
| 17 | *It is not a completely new idea... The authors had better to conclude on the contributions of their methodology as stated in the title.(P25L9-10)* | A new sentence has been added to the conclusion part. | P25L26-28 |

[revised manuscript text omitted]

---

## Author Response (AR3)

Dear Editor,

Please find attached our final version of our paper entitled "On the potential of variational calibration for a fully distributed hydrological model: application on a Mediterranean catchment"

No change has been brought to the content of the paper (including text and figure) since the minor revision iteration. However, we would like to report the following minor changes in our latex code:

1. \documentclass[hessd]{copernicus} to \documentclass[hess]{copernicus}
2. \begin{figure}[] to \begin{figure*}[t]
3. deleted \url{} in the .bib file for the reference (Javelle, P. et al. 2019, Flash flood warnings: Recent achievements in France with the national Vigicrues Flash system., United Nations Office for Disaster Risk Reduction) since the url is not properly splitted for the 2-columns format.

We would like to thank you and the three anonymous reviewers for your constructive comments and suggestions.

Best regards,

On behalf of the authors